# Rethinking Circuit Completeness in Language Models: AND, OR, and ADDER Gates

**Hang Chen**
School of Computer Science and Technology
Xi'an Jiaotong University
albert2123@stu.xjtu.edu.cn

**Jiaying Zhu**
School of Computer Science and Engineering
The Chinese University of Hong Kong
jyzhu24@cse.cuhk.edu.hk

**Xinyu Yang**[*]
School of Computer Science and Technology
Xi'an Jiaotong University
yxyphd@mail.xjtu.edu.cn

**Wenya Wang**[†]
School of Computer Science and Engineering
Nanyang Technological University
wangwy@ntu.edu.sg

## Abstract

Circuit discovery has gradually become one of the prominent methods for mechanistic interpretability, and research on circuit **completeness** has also garnered increasing attention. Methods of circuit discovery that do not guarantee completeness not only result in circuits that are not fixed across different runs but also cause key mechanisms to be omitted. The nature of incompleteness arises from the presence of **OR gates** within the circuit, which are often only partially detected in standard circuit discovery methods. To this end, we systematically introduce three types of logic gates: AND, OR, and ADDER gates, and decompose the circuit into combinations of these logical gates. Through the concept of these gates, we derive the minimum requirements necessary to achieve faithfulness and completeness. Furthermore, we propose a framework that combines noising-based and denoising-based interventions, which can be easily integrated into existing circuit discovery methods without significantly increasing computational complexity. This framework is capable of fully identifying the logic gates and distinguishing them within the circuit. In addition to the extensive experimental validation of the framework's ability to restore the faithfulness, completeness, and sparsity of circuits, using this framework, we uncover fundamental properties of the three logic gates, such as their proportions and contributions to the output, and explore how they behave among the functionalities of language models.

## 1 Introduction

As an intervention-based approach to mechanistic interpretability, circuit discovery allows for the extraction of subgraphs from the computational graph of a language model that play a significant role in task performance, referred to as circuits [1, 2, 3, 4]. Several key studies have supported its development [5, 6], such as those focusing on ensuring that circuits faithfully reflect the model's outputs [2, 7], enabling efficient circuit extraction [8], and addressing scalability challenges for models with extremely large parameters and corpora [9, 10, 11].

As the concept of circuits evolves, recent attention has increasingly focused on the **completeness** of circuits in addition to **faithfulness**. For example, completeness has been redefined such that

---

[*]Corresponding author

[†]Corresponding author

39th Conference on Neural Information Processing Systems (NeurIPS 2025).

when a circuit is removed from the computational graph, the performance of the task should degrade significantly [12, 13]. Nevertheless, current circuit discovery methods have been found to lack completeness [9]. Moreover, theory analysis [14] indicates that incomplete circuits lead to two potential pitfalls: non-transitivity and preemption, which prevent the recovery of the key mechanisms underlying the circuit. Finally, incompleteness results in variability in circuit discovery outcomes, making the circuit appear more like an arithmetic solution to obtain the output rather than representing a closed-form solution with interpretability [15].

Incompleteness largely arises from the presence of **OR gates** [16, 2]. For instance, consider a model $M$ that employs two identical and disjoint serial circuit paths, $C_1$ and $C_2$, which operate in parallel and whose outputs are subsequently combined via an OR operation. In this case, identifying either path is sufficient to achieve faithfulness, and removing the other path is a preferable choice for promoting sparsity. However, restoring completeness by discovering OR gates remains a challenge. The simplest approach, which involves repeated interventions on combinations of components [14], could theoretically uncover the complete OR gate; however, this makes circuit discovery an NP problem (We explained it in Appendix E.1). Additionally, while denoising-based intervention methods can rapidly restore OR gates [17], they lead to a more severe loss of faithfulness. Furthermore, these methods fail to isolate the OR gates from the final circuit, resulting in a lack of logical interpretability.

To this end, we introduce the concept of logic gates, where any circuit can be decomposed into AND, OR, and ADDER gates, and propose a systematic framework to uncover and separate all the gates, and then to explain their correspondence to faithfulness or completeness with the sparsity constraint. Our specific contributions are as follows:

1. **We systematically introduce three types of logic gates that compose a circuit: AND, OR, and ADDER gates.** Through these gates, we are able to infer the minimum requirements for a circuit to achieve faithfulness and completeness, as well as assess the capability of noising-based and denoising-based interventions in restoring these gates. Based on these corollaries, we analyze three types of prevailing circuit discovery methods, named greedy search [2, 18, 11], linear estimation [8, 19], and differentiable mask [9, 12, 10], by evaluating their ability to recover the three logic gates and their faithfulness and completeness. Moreover, we conduct experiments to provide empirical evidence supporting these theoretical conclusions.

2. **We propose a framework capable of fully discovering the three logic gates**, which can be easily extended to current circuit discovery methods with constant-time complexity. Our framework combines noising-based and denoising-based interventions, ensuring both the faithfulness and completeness of the circuit, and enabling the separation of AND, OR, and ADDER gates from the final circuit. Extensive experimental results demonstrate that our framework achieves promising faithfulness and completeness. Additionally, to ensure consistency in the granularity of noising-based and denoising-based interventions, we introduce a misalignment score for AND and OR gates to measure whether the scales of the two intervention strategies are aligned when combined.

3. **We explore the characteristics of AND, OR, and ADDER gates in a circuit**, including their proportions and contributions to the output, building upon our proposed logic gates and recovery framework. Furthermore, we examine the relationship between logic gates and the functionality of language models. Experimental results show that OR gates typically link multiple backup paths for the same function, while AND gates often connect paths for different necessary functions.

## 2 Preliminaries

### 2.1 Circuit Discovery

In Transformer decoder-based language models, the forward pass is typically conceptualized as a **computational graph** $\mathcal{G}$, where the nodes represent components (such as attention heads, MLPs, or even more granular elements like the query, key, and value matrices) and an edge $i \to j$ denotes a connection where the output of component $i$ serves as input to component $j$. Circuit discovery seeks to identify a subgraph (circuit) $\mathcal{C} \subset \mathcal{G}$ that captures the task-relevant behavior of the model [3].

The process used to prune and obtain the circuit $\mathcal{C}$ is referred to as **intervention** (also known as **knockout**, **ablation**) [17, 20, 21, 22]. For a given task $\mathcal{T}$, each sample $x$ is referred to as **clean text**, and the corresponding forward pass yields the **clean activation** $x_i$ at each component $i$. A perturbed version of the input, denoted $\tilde{x}$, is called **corrupted text**, producing a corresponding **corrupted**

**activation** [23, 17]. The corrupted activation $\tilde{x}_i$ depends on the specific ablation method used. For example, ZERO ABLATION sets $\tilde{x}_i = 0$, while NOISE ABLATION draws $\tilde{x}_i$ from a predefined noise distribution. A widely used method, INTERCHANGE ABLATION, defines $\tilde{x}_i$ as the activation resulting from an input text that has been minimally perturbed to produce a different task label [10].

The intervention is divided into two strategies: **noising-based intervention** (hereafter referred to as **Ns**) and **denoising-based intervention** (hereafter referred to as **Dn**) [24]. The **Ns** first runs the clean text in the computational graph. Then, corrupted activations replace each clean activation to observe the change in the final output $y$. If replaced (also known as removed or pruned) activations lead to a significant change in output, they are considered to make an important contribution to the task $\mathcal{T}$ and should be retained in the circuit $\mathcal{C}$ [17]. Let $p_{\mathcal{G}}(y|x)$ denote the model's original output, $p_{\mathcal{C}}(y|x, \tilde{x})$ represent the circuit's output after intervention. Specifically, if an edge $j \rightarrow i$ is retained within $\mathcal{C}$, the activation of component $i$ keeps the clean one ($x_i$). Conversely, it is replaced by the corrupted one ($\tilde{x}_i$). Let $s$ denote the requirement of sparsity, and $D$ represent the distance used to quantify the difference between the two outputs. Ns has the following objective:

$$\arg\min_{\mathcal{C}} \mathbb{E}_{(x,\tilde{x}) \in \mathcal{T}}[D(p_{\mathcal{G}}(y|x)||p_{\mathcal{C}}(y|x, \tilde{x}))], \;\; s.t. \; 1 - |\mathcal{C}|/|\mathcal{G}| \geq s \tag{1}$$

Equation 1 indicates that the circuit is a subgraph that most closely approximates the functionality of the computational graph, where the components and edges have the most significant effect on the output. Similarly, the **Dn** first performs the corrupted run in the computational graph, and then replaces the corrupted activations with the clean activations. Those activations that lead to significant changes in the output ($\tilde{y}$) consist of the circuits. Dn thus has the following objective:

$$\arg\min_{\mathcal{C}} \mathbb{E}_{(x,\tilde{x}) \in \mathcal{T}}[D(p_{\mathcal{G}}(\tilde{y}|\tilde{x})||p_{\mathcal{C}}(\tilde{y}|\tilde{x}, x))], \;\; s.t. \; 1 - |\mathcal{C}|/|\mathcal{G}| \geq s \tag{2}$$

Most of the related work on circuit discovery follows the **Ns** strategy. We categorize these works into three types: (1) **Greedy search** [2, 18, 11], which iteratively examines each edge (or node) through intervention to obtain a greedy solution for the circuit. (2) **Linear estimation** [8, 19], where the contribution of each edge is approximated by a gradient measure obtainable in a single backward pass. This approach ranks the importance of each edge to approximate the circuit. (3) **Differentiable masks** [9, 12, 10], where a learnable mask is assigned to each edge (or node), treating circuit discovery as an optimization problem to derive the optimal circuit.

## 2.2 Circuit Evaluation

Circuit evaluation is primarily defined by three aspects: **faithfulness**, **completeness**, and **sparsity**.

**Faithfulness** refers to the circuit's ability to perform task $\mathcal{T}$ in isolation, which is defined as the difference between the circuit's output and the model's original output [16, 9, 17]. This is represented in Equations 1 as $\mathbb{E}_{(x,\tilde{x}) \in \mathcal{T}}[D(p_{\mathcal{G}}(y|x)||p_{\mathcal{C}}(y|x, \tilde{x}))]$ (simplified as $D(\mathcal{G}||\mathcal{C})$). Method ACDC [16] measures faithfulness by computing the average difference in the unnormalized output logits between the correct token and an incorrect option. Recently, work [2, 17, 25] proposes that KL divergence provides a better measure of the distribution over the vocabulary, while other work [9, 15] suggests that task accuracy can avoid the overemphasis on irrelevant vocabulary in the KL divergence. In this paper, we measure faithfulness using both KL divergence and task accuracy as metrics.

**Completeness** refers to whether the circuit includes all the important paths that have an effect on the output. The work [16] first introduces the concept of circuit completeness, stating that $\mathcal{C}$ and $\mathcal{G}$ should ensure similar outputs even under any knockout. Therefore, the incompleteness score is defined as the difference $D(\mathcal{C} \setminus \mathcal{K}||\mathcal{G} \setminus \mathcal{K})$ for any subcircuit $\mathcal{K} \subset \mathcal{C}$. Existing work [9, 15] proposes that insufficient sampling of $\mathcal{K}$ may lead to unreliable approximations (we show the practical results in Appendix A), and thus recommends evaluating completeness by assessing the performance after the circuit's removal from the computational graph on the task $\mathcal{T}$, i.e., $D(\mathcal{G} \setminus \mathcal{C}||\mathcal{G})$ [13, 12]. In this paper, we also adopt it to evaluate completeness.

**Sparsity** refers to that the circuit should be as small as possible. Currently, many works [10, 9, 15] recommend measuring sparsity using the ratio $|\mathcal{C}|/|\mathcal{G}|$, which represents the proportion of edges in the circuit relative to those in the computational graph. In fact, higher sparsity tends to result in lower faithfulness, meaning that the circuit always reflects some trade-off between sparsity and faithfulness.

# 3 Circuit Logic

## 3.1 Logical Gates

Recent studies [16, 2, 17] have increasingly observed that within a circuit, certain subcircuits influence the output according to logical relationships resembling AND or even OR operations. Building on these findings, we systematically introduce three fundamental circuit logic types: the **AND** gate, **OR** gate, and **ADDER** gate.

**Definition 1.** *We assume a common paradigm in which a receiver node $B$, which is connected by more than 1 sender node $A_1, A_2, \ldots$. For any edge $A_i \to B$, we use binary values '0' and '1' to represent the activation state of a node. Specifically, $A_i = 0$ indicates that node $A_i$ is removed, ablated, or deactivated, whereas $A_i = 1$ indicates that node $A_i$ is retained and active. When the sender nodes are ablated, the effect of node $B$ on the output exhibits three distinct patterns, which are as follows:*

*AND: All sender nodes satisfy an AND logical relationship with the receiver node, i.e., $B = A_1 \wedge A_2 \wedge \ldots$. In this case, node $B$ exerts a significant effect on the output only if all of its sender nodes are retained. If even a single sender node is ablated, the effect of $B$ on the output is nearly eliminated.*

*OR gate: All sender nodes satisfy an OR logical relationship with the receiver node, i.e., $B = A_1 \vee A_2 \vee \ldots$. In this case, node $B$ always exerts a significant effect on the output if one or more of its sender nodes are retained. Only if all sender nodes are ablated, the effect of $B$ on the output is nearly eliminated.*

*ADDER gate: all sender nodes satisfy an ADDER logical relationship with the receiver node, i.e., $B = A_1 + A_2 + \ldots$. In this case, node $B$ exhibits its maximal effect on the output only when all of its sender nodes are retained. If any single sender node is ablated, the effect of $B$ on the output is substantially diminished; when all sender nodes are ablated, $B$'s effect on the output is reduced to zero. Accordingly, we define the state of $B$ as taking values $0,1,2,\ldots$, where the total number of distinct states equals the number of sender nodes.*

Theoretical analyses support the view that Ns is capable of recovering a complete AND gate but fails to recover a complete OR gate, whereas Dn demonstrates the opposite pattern [17]. This asymmetry is straightforward to interpret. The Ns procedure corresponds to the transition from a clean activation state (state $= 1$) to a corrupted activation state (state $= 0$). Since all gates can be regarded as being initialized with activation states equal to 1, any transition to state $= 0$ induces a significant change in the effect of AND and ADDER gates on the output. Consequently, Ns can reliably identify AND and ADDER gates. Conversely, the Dn procedure corresponds to initialization with activation states equal to 0. In this case, any transition to state $= 1$ produces a significant change in the effect of OR and ADDER gates on the output. Moreover, we design a toy model to explain ADD, OR, ADDER gates in Appendix D.

Therefore, we denote the circuit constructed under the Ns strategy as $\mathcal{C}_{\text{Ns}}$, and the one constructed under the Dn strategy as $\mathcal{C}_{\text{Dn}}$. Based on the above set-theoretic relationships between $\mathcal{C}_{\text{Ns}}$ and $\mathcal{C}_{\text{Dn}}$, we extract subsets of edges corresponding to AND, OR, and ADDER gates as follows:

- AND gate ($\mathcal{C}_{\text{AND}}$): edges that are present in $\mathcal{C}_{\text{Ns}}$ but absent from $\mathcal{C}_{\text{Dn}}$.
- OR gate ($\mathcal{C}_{\text{OR}}$): edges that are present in $\mathcal{C}_{\text{Dn}}$ but absent from $\mathcal{C}_{\text{Ns}}$.
- ADDER gate ($\mathcal{C}_{\text{ADDER}}$): edges that are shared between $\mathcal{C}_{\text{Ns}}$ and $\mathcal{C}_{\text{Dn}}$.

We conduct an ablation on these edges: for each gate, we randomly remove either one or two edges on the same receiver node and measure the resulting change in the KL divergence of the output. This procedure is repeated 30 times for each receiver node, and the distributions of $\Delta$KL values are summarized via box plots, as shown in Figure 1 (Detailed results are shown in Appendix E.2). We selected the computational graph of GPT2-small as $\mathcal{G}$, and Indirect Object Inference (IOI) [16] as the test task. For the baseline methods, we chose ACDC [2] to represent the greedy search method, EAP [8] to represent the linear estimation method, and EdgePruning [10] to represent the differentiable mask method. For details regarding the implementation of these strategies within each baseline, we refer the reader to Appendix E.

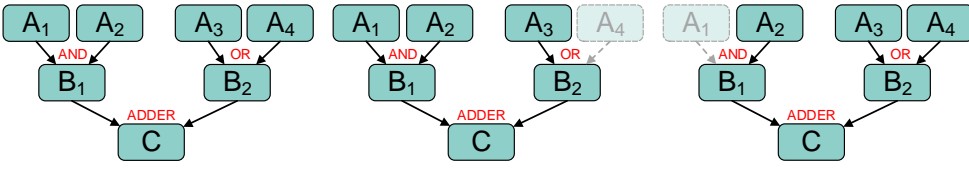

(a) Complete Circuit     (b) Max faithfulness and sparsity (c) Max completeness and sparsity

Figure 2: Presentation of a toy model designed to elucidate the logical relationships among faithfulness, completeness, and sparsity. Suppose that $A_1 \wedge A_2 = B_1$, $A_3 \vee A_4 = B_2$, and $B_1 + B_2 = C$, and among the three only $C$ is connected to the output. When optimizing for faithfulness and sparsity alone, it is possible to remove one edge from the OR gate (either $A_3 \to B_2$ or $A_4 \to B_2$), thereby ensuring the minimal number of edges. Similarly, when optimizing for completeness and sparsity, one edge from the AND gate (either $A_1 \to B_1$ or $A_2 \to B_1$) can be eliminated for sparsity.

In Figure 1, for **AND** gates, the $\Delta$KL values resulting from removing one versus two edges are similar, consistent with the conclusion that the disruption of any single edge in AND gates renders the gate ineffective. For **OR** gates, removing a single edge has little effect on $\Delta$KL, supporting the idea that the OR gates remains functional as long as at least one edge in OR gates remains intact. In contrast, for **ADDER** gates, removing two edges leads to a significantly larger increase in $\Delta$KL compared to removing one, indicating that the edges contribute independently to the gate's function.

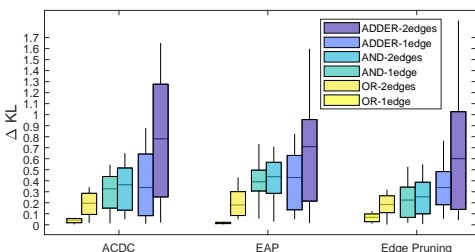

Figure 1: $\Delta KL$ in removing 1 and 2 edges of AND, OR, ADDER gates.

### 3.2 How to Use Logical Gates to Interpret Faithfulness and Completeness

These logical gates reveal some interesting phenomena as shown in Figure 2. For optimal faithfulness and sparsity, the circuit only needs to include **one** edge from each OR gate. For optimal completeness and sparsity, the circuit only needs to include **one** edge from each AND gate. Based on the definitions in Section 2, we can draw the following corollary regarding these properties:

**Corollary 1.** *The minimal edge subset that satisfies optimal faithfulness consists of **all** edges from the AND gates, **all** edges from the ADDER gates, and any **one** edge from each OR gate. The minimal edge subset that satisfies optimal completeness consists of **all** edges from the OR gates, **all** edges from the ADDER gates, and any **one** edge from each AND gate (The proofs are shown in Appendix C).*

Corollary 1 provides an explanation for why existing circuit discovery algorithms predominantly adopt the Ns strategy. The reason is that Ns is able to recover complete AND and ADDER gates, while its recovery of OR gates remains incomplete. This tradeoff corresponds precisely to the optimal balance between faithfulness and sparsity. Conversely, the Dn strategy is not adopted because it fails to recover complete AND gates, thereby severely compromising the faithfulness of the resulting circuit.

Therefore, we further evaluate the performance of existing work in terms of faithfulness and completeness. Table 1 presents the specific results for the three types of circuit discovery methods mentioned in Section 2.

Among these, methods based on greedy search and differentiable masks can identify partial OR gates, whereas methods based on linear estimation are unable to detect any edges of OR gates. Similarly, methods from Dn exhibit a similar pattern. While they can completely identify OR and ADDER gates, methods based on greedy search and differentiable masks can detect partial AND gates, while methods based on linear estimation fail to identify any. In Appendix D, we explain why greedy search and differentiable mask methods are able to identify some edges, whereas linear estimation completely fails to do so. Moreover, inspired by [2], we design a simple one-layer transformer toy model to implement the basic AND, OR, and ADDER gates, and validate the performance of these circuit discovery methods corresponding to the conclusion from Table 1.

Table 1: Capabilities and performances of three types of circuit discovery methods in recovering logical gates, faithfulness, and completeness. The symbol $\sqrt{}$ represents the ability to fully satisfy the corresponding requirement, $\times$ indicates the complete inability to satisfy the corresponding requirement, and $\bigcirc$ denotes the ability to partially satisfy the corresponding requirement.

| Strategy | Method | AND | OR | ADDER | Faithfulness | Completeness |
|----------|--------|-----|-----|-------|--------------|--------------|
| Ns | greedy search [2, 18, 11] | $\sqrt{}$ | $\bigcirc$ | $\sqrt{}$ | $\sqrt{}$ | $\times$ |
|  | linear estimation [8, 19] | $\sqrt{}$ | $\times$ | $\sqrt{}$ | $\times$ | $\times$ |
|  | differentiable mask [9, 12, 10] | $\sqrt{}$ | $\bigcirc$ | $\sqrt{}$ | $\sqrt{}$ | $\times$ |
| Dn | greedy search [2, 18, 11] | $\bigcirc$ | $\sqrt{}$ | $\sqrt{}$ | $\times$ | $\sqrt{}$ |
|  | linear estimation [8, 19] | $\times$ | $\sqrt{}$ | $\sqrt{}$ | $\times$ | $\times$ |
|  | differentiable mask [9, 12, 10] | $\bigcirc$ | $\sqrt{}$ | $\sqrt{}$ | $\times$ | $\sqrt{}$ |

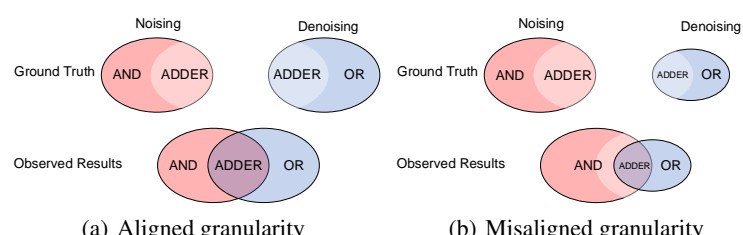

(a) Aligned granularity          (b) Misaligned granularity

Figure 3: A Venn diagram for $\mathcal{C}_{\text{Ns}}$ and $\mathcal{C}_{\text{Dn}}$. In the case of granularity alignment, the intersection correctly separates the AND, OR, and ADDER gates (left figure). However, in the case of misalignment, it results in some ADDER gates being incorrectly classified as AND (or OR) gates (right figure).

### 3.3 Granularity Alignment between $\mathcal{C}_{\text{Ns}}$ and $\mathcal{C}_{\text{Dn}}$

The intersection operation mentioned above raise concerns about the **granularity alignment** between $\mathcal{C}_{\text{Ns}}$ and $\mathcal{C}_{\text{Dn}}$. As illustrated in Figure 3, if the number of edges in $\mathcal{C}_{\text{Ns}}$ significantly exceeds that in $\mathcal{C}_{\text{Dn}}$, some edges identified as the AND gates could belong to the true type of the ADDER gates. This misalignment in granularity can also occur when $\mathcal{C}_{\text{Dn}}$ is considerably larger. Therefore, we propose two metrics (refer to Appendix F) to assess the degree of misalignment between $\mathcal{C}_{\text{Ns}}$ and $\mathcal{C}_{\text{Dn}}$ when performing intersection.

We report the misalignment of $\mathcal{C}_{\text{Ns}}$ and $\mathcal{C}_{\text{Dn}}$ at different scales in Appendix F. The results indicate that when the number of edges in the Dn circuit is approximately equal to that in the Ns circuit, both the misalignment score and its standard deviation reach an acceptable level. Therefore, throughout this paper, we assume that the optimal alignment occurs when Ns and Dn contain an **equal number of edges** and conduct experiments based on this assumption by scaling the number of edges identified by Ns and Dn strategies in a similar range.

## 4 Discovering Logically Sound Circuit

### 4.1 Optimization for Logically Sound Circuit

Existing baseline methods are capable of recovering only complete AND and ADDER structures, as demonstrated in Table 1. Therefore, the recovery of complete OR gates remains a challenge. Several approaches can be considered to address this problem, such as introducing additional combinations of interventions or varying the order of the intervention to identify different surviving edges of the OR gate, or incorporating a completeness score, such as $D(\mathcal{G} \setminus \mathcal{C} \,\|\, \mathcal{G})$, into the circuit discovery process. However, these approaches come with significant drawbacks. Expanding the space of intervention combinations renders circuit discovery an NP problem. Meanwhile, the inclusion of completeness scores is incompatible with non-differentiable optimization strategies such as greedy search, and it also fails to effectively split the three logic gate types in the recovered circuits.

Therefore, we propose a combined **Ns+Dn** approach to recover logically complete gates. This method is compatible with a wide range of circuit discovery algorithms, introduces minimal additional computational overhead, and enables clear and effective separation of the three types of logic gates.

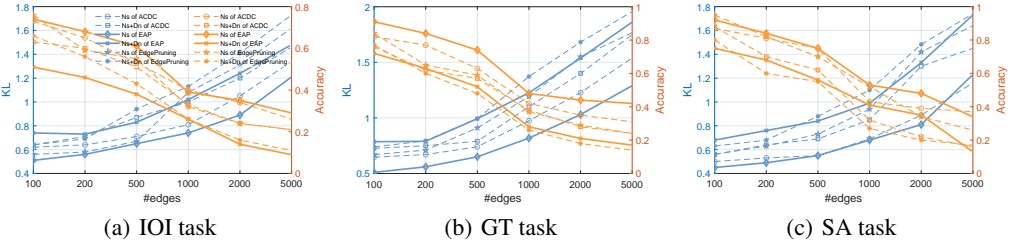

|       | (a) IOI task | (b) GT task | (c) SA task |
|-------|:------------:|:-----------:|:-----------:|

Figure 4: Completeness evaluation of circuit from Ns and NS+Dn.

Ns+Dn has the following objective:

$$\arg\min_{\mathcal{C}} \mathbb{E}_{(x,\tilde{x})\in\mathcal{T}}[D(p_{\mathcal{G}}(y|x)||p_{\mathcal{C}}(y|x,\tilde{x})) + D(p_{\mathcal{G}}(\tilde{y}|\tilde{x})||p_{\mathcal{C}}(\tilde{y}|\tilde{x},x))], \ \ s.t. \ 1-|\mathcal{C}|/|\mathcal{G}| \geq s \quad (3)$$

In brief, for each baseline, we modify its implementation to perform both the Ns and Dn strategies in parallel, whereas originally only the Ns strategy was applied.

## 4.2 Validation of Logically Sound Circuit

In this subsection, we focus on the **faithfulness** and **completeness** of the logically sound circuit (from our framework in Section 4.1, denoted by $\mathcal{C}_{\text{Ns+Dn}}$) and the circuit of existing work (since existing work generally adopts Ns as a basic intervention strategy, we denote it by $\mathcal{C}_{\text{Ns}}$). Similar to Section 3.1, we select GPT2-small as the computational graph, and ACDC, EAP, and EdgePruning as methods to represent greedy search, linear estimation, and differentiable mask, respectively. We examine the circuits obtained through Ns, Dn, and Ns+Dn. For instance, in ACDC, when intervening on each edge, we simultaneously compute the effect of substituting the clean activation with a corrupted one in the clean run, and the effect of substituting the corrupted activation with a clean one in the corrupted run. In EAP, we compute gradients under both clean and corrupted conditions. For EdgePruning, we replace Equation 1 with Equation 3 as the optimization objective. Detailed implementation can be found in Appendix E.3. These experiments are conducted on three mainstream tasks for circuit discovery, namely indirect object inference (IOI) [16], greater than (GT) [26], and syntactic agreement [9]. The details of these tasks are presented in Table 2.

Table 2: An overview of the tasks and datasets.

| Task | Example([Corrupted text]) | Output | corrupted output |
|------|---------------------------|--------|------------------|
| IOI  | When Mary and John went to the store, John (Alice) gave a drink to | Mary | other names |
| GT   | The war lasted from 1517 (1501) to 15 | 18 or 19 or... 99 | other digits |
| SA   | Many girls (girl) insulted | themselves | herself |

### 4.2.1 Completeness

Following the definition of completeness in Section 2, we first compare the changes in KL divergence and accuracy for the corresponding tasks (IOI, GT, SA) when the circuit is removed from the computational graph. Specifically, we compare the differences between the original circuits (obtained through Ns) and the logically sound circuits (obtained through Ns+Dn) after removal, for three methods: ACDC, EAP, and EdgePruning. To account for the effects of sparsity, we constrain the number of edges in both circuits to remain consistent across six sparsity levels: 100, 200, 500, 1000, 2000, and 5000 edges. Figure 4 shows that, both in terms of KL divergence and accuracy, the performance of circuits removed through Ns+Dn is noticeably weaker compared to those removed through Ns. This corroborates Corollary 1, where we note that Ns, due to its inability to fully recover the OR gate, results in suboptimal completeness. Additionally, we observe that the gap between Ns and Ns+Dn is largest in both metrics in the EAP method (see the solid line in Figure 4), where Ns fails to recover any edges of the OR gate. Additionally, since ACDC and EdgePruning are generally able to identify one OR edge, the recovered OR edge exhibits some degree of randomness. In the case of ACDC, this randomness is influenced by the search order, while in the case of EdgePruning, it is influenced by the initial values of the mask. Moreover, we show the detailed results including Dn in Appendix G.

Table 3: Difference in Hamming distance between the $\mathcal{C}_{Ns}$ and $\mathcal{C}_{Ns+Dn}$ (we compute the average Hamming distance between $\mathcal{C}_{Ns}$ and subtract the average Hamming distance between $\mathcal{C}_{Ns+Dn}$). A larger value indicates that the circuits obtained through Ns exhibit greater randomness compared to those obtained through Ns+Dn. #edges represents the number of edges in circuits.

| #edges | IOI | | | GT | | | SA | | |
|---|---|---|---|---|---|---|---|---|---|
| | ACDC | EAP | EdgePruning | ACDC | EAP | EdgePruning | ACDC | EAP | EdgePruning |
| 100 | 3.4±0.6 | 0.6±0.1 | 8.4±3.7 | 4.8±0.9 | 0.5±0.1 | 12.7±4.9 | 2.8±0.4 | 1.1±0.2 | 15.3±5.8 |
| 200 | 5.9±1.3 | 1.2±0.3 | 18.1±6.7 | 6.7±1.8 | 1.3±0.2 | 22.5±9.1 | 4.3±0.9 | 2.2±0.5 | 28.4±12.7 |
| 500 | 14.7±3.7 | 1.8±0.7 | 44.5±13.8 | 16.9±4.2 | 1.6±0.8 | 49.1±15.6 | 12.8±2.9 | 2.9±0.9 | 55.9±16.7 |
| 1000 | 21.8±5.3 | 4.7±1.8 | 89.6±27.9 | 23.6±6.4 | 4.4±1.6 | 97.5±29.4 | 19.7±4.3 | 5.7±2.8 | 108.2±31.4 |
| 2000 | 49.5±12.9 | 7.9±2.9 | 195.3±57.8 | 55.7±14.9 | 8.6±3.5 | 211.7±66.2 | 44.8±15.2 | 8.8±3.1 | 237.4±64.8 |
| 5000 | 127±28.5 | 14.5±6.9 | 509.5±164.7 | 136.5±33.4 | 15.9±6.1 | 564.8±181.1 | 113.7±5.8 | 15.4±5.8 | 688.9±144.5 |

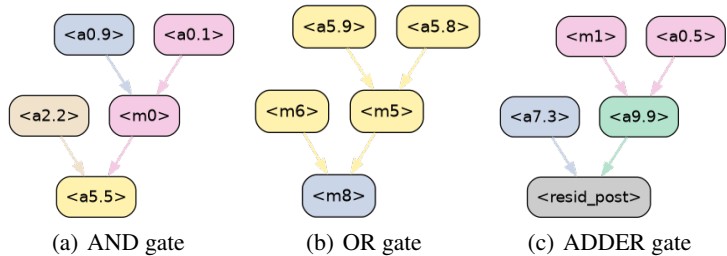

(a) AND gate        (b) OR gate        (c) ADDER gate

Figure 6: The cases with 2-layer gates of AND, OR, ADDER circuits.

To further validate completeness, we test the overlap of randomly generated circuits by extracting 30 distinct circuits under different random seeds and calculating the Hamming distance between pairs of these circuits to assess the randomness of the discovered circuits. Table 3 shows that the randomness of ACDC and EdgePruning is significantly higher than that of EAP, and it increases with the sparsity scales (as more OR gates are discovered). Additionally, the randomness of the circuits obtained through Ns+Dn is consistently lower than that of the circuits obtained through Ns, further supporting the claim that the inclusion of all three logical gates ensures optimal completeness.

### 4.2.2 Faithfulness

In Appendix H, we compare the circuits obtained using three strategies—Ns, Dn, and Ns+Dn—under the same sparsity constraints (specifically, we select edge counts of 100, 200, 500, 1000, 2000, and 5000) in terms of KL divergence and accuracy. The results show that, in terms of faithfulness, we have the relationship: $\mathcal{C}_{Ns+Dn} \approx \mathcal{C}_{Ns} > \mathcal{C}_{Dn}$. Figure 5 illustrates the average of the three methods on the IOI task to corroborate this conclusion. More results can be found in Figure 9 (a)-(c), which further supports the faithfulness requirements asserted in Corollary 1.

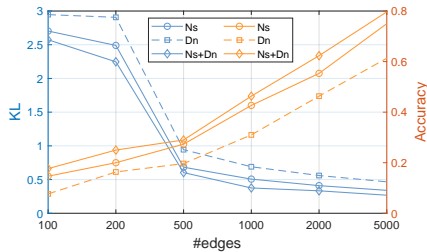

Figure 5: The average of three methods in faithfulness of IOI task.

## 5 Exploration on Logical Gates

### 5.1 Graph Study

In Appendix I, we present the circuits of the AND, OR, and ADDER gates recovered by ACDC on the IOI task and map the functions of the components to those in the previous IOI circuit [16]. We show parts of these circuits as demonstrations in Figure 6; each color represents one function in IOI circuit and blocks represent components at different locations, such as "a5.9" indicating the 9-th attention head in the 5-th layer, and "m8" referring to the MLP in the 8-th layer. The complete gate circuit can be found in Figure 10 of Appendix I. The results are interesting, revealing that the **AND gates typically receive edges from different functions**, suggesting that these functions must work together to support the receiver's activation. In contrast, **the OR gates almost exclusively receive edges from the same function**, indicating that these edges are likely interchangeable due to their

execution of the same function. **The ADDER gates, on the other hand, tend to focus on combining two functions from different layers**, with the activation generally considering the outputs of both shallow-layer and deep-layer functions.

## 5.2 Output Contribution

In Appendix J, we investigate the contribution of three types of logic gates to the output. The gate effect represents the contribution of the entire gate to the output and the edge effect represents the average contribution of each edge to the output. The results show that the contribution of the ADDER gates is significantly higher than that of the AND and OR gates. Furthermore, methods that focus on the edge effect, such as differentiable masks and linear estimation, lead to a higher average effect in the recovered circuit.

## 5.3 Proportion

In Appendix K, we present the number of AND, OR, and ADDER edges recovered by different methods. **The results indicate that the proportion is closely related to the type of circuit discovery method used.** For instance, greedy search selects all edges beyond the threshold, resulting in nearly equal numbers of AND, OR, and ADDER edges. In contrast, differentiable mask methods calculate the effect of each edge, which is disadvantageous for gates like AND and OR that contain multiple edges. As a result, the number of ADDER edges is significantly higher.

# 6 Conclusions

This paper systematically introduces three logic gates—AND, OR, and ADDER—to explain the essential requirements of circuit faithfulness and completeness. Furthermore, it provides an analysis of how existing circuit discovery methods perform with respect to these logic gates. Additionally, we propose an Ns&Dn-based method for separating the three logic gates, and for restoring a logically sound circuit. We empirically validate the differences in faithfulness and completeness between the logically sound circuit and existing circuits. Finally, we explore the relationships between the logic gates in terms of distribution, contribution, and functionality.

## 6.1 Limitations and Future Research

The three logical gates under discussion in this paper are, in principle, derived from the intersection of $\mathcal{C}_{Ns}$ and $\mathcal{C}_{Dn}$. Nevertheless, it must be acknowledged that other types of logical gates may also exist—for instance, the XOR gate—whose underlying logic cannot be captured solely through the simple intersection of $\mathcal{C}_{Ns}$ and $\mathcal{C}_{Dn}$. That said, we contend that the three gates obtained from this intersection—AND, OR, and ADDER—are already sufficient to cover all edges of the circuits constructed from $\mathcal{C}_{Ns}$ and $\mathcal{C}_{Dn}$. For the purposes of the present scope of research, this coverage is adequate.

Moreover, with a complete understanding of the logical relationships between edges, the circuit becomes more useful for offering insights into model control. For instance, a logically sound circuit offers a novel approach for verifying the potential combination of tasks through boolean satisfiability, which is treated as our future study. We have demonstrated the potential contributions of completeness research with a toy task of model unlearning, as shown in Appendix L.

## 6.2 Societal and Ethical Impact

Our work aims to facilitate the process of understanding and explaining the logical connections in language models, which is crucial for their continued safe development and deployment. We do not foresee logically sound circuit and logical gates being used towards adverse societal or ethical ends.

## Acknowledgments and Disclosure of Funding

This research/project is supported by the National Research Foundation, Singapore under its National Large Language Models Funding Initiative, (AISG Award No: AISG-NMLP-2024-005).

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

# A  The Preference between Two Types of Completeness Metrics

The earlier completeness evaluation method (proposed by [16]) in fact measures *incompleteness*, defined as $D(C \setminus K \,\|\, G \setminus K)$, where a lower score indicates better performance. In contrast, the current method [9] measures *completeness*, defined as $D(G \setminus C \,\|\, G)$, where a higher score is preferable. Importantly, these two metrics are not directly interchangeable — that is, $1 - D(C \setminus K \,\|\, G \setminus K) \neq D(G \setminus C \,\|\, G)$.

Nevertheless, a comparison can still be made in terms of the **standard deviation**. It has been theoretically demonstrated that the earlier completeness method suffers from high variance due to insufficient sampling, leading to instability in the final results. To illustrate this, we provide a simple comparison between the previous and current evaluation methods, as shown in the Table 4:

Table 4: The standard deviation between two metrics of completeness.

| Method | 5 sampled K | 10 sampled K | 20 sampled K | 30 sampled K | New metric |
|--------|-------------|--------------|--------------|--------------|------------|
| ACDC   | 1.1±1.6     | 1.8±1.2      | 1.4±1.1      | 1.2±0.6      | 1.3±0.13   |
| EAP    | 1.7±1.5     | 1.3±1.6      | 1.6±1.2      | 1.3±0.7      | 1.4±0.11   |
| Edge-P | 1.5±1.8     | 1.4±1.9      | 1.4±1.6      | 1.6±0.6      | 1.2±0.08   |
| Ours   | 0.4±0.3     | 0.6±0.4      | 0.3±0.3      | 0.3±0.2      | 1.8±0.03   |

We conducted five different runs of circuit discovery using random seeds, with the random subset $K$ chosen from subcircuits of sizes ranging from 2 to 5 nodes. For the previous evaluation metric, we performed 5, 10, 20, and 30 sampling iterations.

The experimental results clearly support two conclusions:

1. For algorithms that inherently lack completeness (such as ACDC, EAP, and Edge-Pruning), as well as for our own method, which ensures completeness, the previous evaluation metric exhibits large standard deviations across all trials. This indicates that the sampling process for the previous metric is insufficient, rendering the results highly unreliable.

2. The new evaluation metric significantly reduces variance for all types of circuits, as it does not suffer from sampling issues. The variance observed in the results arises from minor differences in the circuits generated by different random seeds.

# B  How Does Circuit Logic Model Intervention?

## B.1  AND Gate

Consider a simple logical gate: $A_1 \wedge A_2 = B$ (only if both $x_{A_1}$ and $x_{A_2}$ are present can $B$ be activated), where $B$ directly influences the output. For noising-based intervention (**Ns**), replacing $x_{A_1}$ with $\tilde{x}_{A_1}$, or $x_{A_2}$ with $\tilde{x}_{A_2}$, produces a significant effect on the output. Thus, Ns is capable of detecting the structure $\{(A_1, A_2), B\}$.

However, for denoising-based intervention (**Dn**), substituting $\tilde{x}_{A_1}$ with $x_{A_1}$ alone does not yield a noticeable change in the output, as $\tilde{x}_{A_2}$ remains present. Similarly, replacing $\tilde{x}_{A_2}$ with $x_{A_2}$ while $\tilde{x}_{A_1}$ is still active also fails to significantly affect the output.

Under a greedy search strategy, if $\tilde{x}_{A_1}$ is first replaced by $x_{A_1}$ and the output remains unchanged, the algorithm concludes that $A_1$ is not relevant and removes it (i.e., replaces $\tilde{x}_{A_1}$ with $x_{A_1}$ in the scenario of Dn). Subsequently, replacing $\tilde{x}_{A_2}$ with $x_{A_2}$ causes a substantial shift in the output due to the presence of $x_{A_1}$, thereby restoring the structure $\{(A_2), B\}$, since $A_1$ has already been removed.

Analogously, a greedy search that begins with $A_2$ and then proceeds to $A_1$ would only recover the structure $\{(A_1), B\}$. Therefore, we conclude that **Ns is capable of identifying the complete AND gate structure, whereas Dn either fails to detect the AND gates or only partially recovers it under greedy search conditions.**

### B.2 OR Gate

Consider a simple logical gate: $A_1 \vee A_2 = B$ (if either $x_{A_1}$ or $x_{A_2}$ are present can $B$ be activated), where $B$ directly influences the output. For **Ns**, replacing $x_{A_1}$ with $\tilde{x}_{A_1}$ alone does not yield a noticeable change in the output, as $x_{A_2}$ remains present. Similarly, replacing $x_{A_2}$ with $\tilde{x}_{A_2}$ while $x_{A_1}$ is still active also fails to significantly affect the output.

Under a greedy search strategy, if $x_{A_1}$ is first replaced by $\tilde{x}_{A_1}$ and the output remains unchanged, the algorithm concludes that $A_1$ is not relevant and removes it (i.e., retains $\tilde{x}_{A_1}$). Subsequently, replacing $x_{A_2}$ with $\tilde{x}_{A_2}$ causes a substantial shift in the output due to the lack of support of $x_{A_1}$, thereby restoring the structure $\{(A_2), B\}$, since $A_1$ has already been removed.

Analogously, a greedy search that begins with $A_2$ and then proceeds to $A_1$ would only recover the structure $\{(A_1), B\}$.

However, for **Dn**, replacing $\tilde{x}_{A_1}$ with $x_{A_1}$, or $\tilde{x}_{A_2}$ with $x_{A_2}$, produces a significant effect on the output. Thus, Dn is capable of detecting the structure $\{(A_1, A_2), B\}$.

Therefore, we conclude that **Dn is capable of identifying the complete OR gate structure, whereas Ns either fails to detect the OR gates or only partially recovers it under greedy search conditions.**

### B.3 ADDER Gate

Consider a simple logical gate: $A_1 + A_2 = B$, where $B$ directly influences the output. Since the influence of each edge in an ADDER gate is independent, removing an edge in either Ns or Dn directly impacts the output via its effect on $B$. For instance, in Ns, replacing $x_{A_1}$ with $\tilde{x}_{A_1}$ results in $B^* = A_2$, which is significantly smaller than $B = A_1 + A_2$. Similarly, in Dn, replacing $\tilde{x}_{A_1}$ with $x_{A_1}$ yields $B^* = A_1$, which is substantially greater than $B = 0$. **Therefore, both Ns and Dn are capable of identifying the complete structure of the ADDER gate.**

## C How Does Circuit Logic Affect Faithfulness, Completeness, and Sparsity?

### C.1 Faithfulness

As introduced in Section 2, faithfulness requires that $D(G||C)$ be minimized. Let us consider the following scenarios:

- For any gate $\{(A_1, A_2), B\}$, if the circuit does not include all edges or nodes from this gate, it is always possible to find a circuit $C^* = C \cup A_1, A_2, B$ such that $D(G||C) > D(G||C^*)$.

- For an AND gate $\{(A_1, A_2), B\}$, if the circuit $C$ only includes $A_1$ and $B$, the gate effect of this AND gate is not maximized (the influence of $B$ is maximized when both $A_1$ and $A_2$ are present). Therefore, it is always possible to find a circuit $C^* = C \cup A_2$ such that $D(G||C) > D(G||C^*)$.

- For an ADDER gate $\{(A_1, A_2), B\}$, if the circuit $C$ only includes $A_1$ and $B$, the gate effect of this ADDER gate is not maximized (again, the influence of $B$ is maximized when both $A_1$ and $A_2$ are present). Thus, there exists a circuit $C^* = C \cup A_2$ such that $D(G||C) > D(G||C^*)$.

- For an OR gate $\{(A_1, A_2), B\}$, if the circuit $C$ only includes $A_1$ and $B$, the gate effect of this OR gate is already maximized (the same applies if only $A_2$ and $B$ are included). Therefore, for $C^* = C \cup A_2$, we have $D(G||C) = D(G||C^*)$. However, from the perspective of sparsity, $|C^*| > |C|$.

Thus, to achieve optimal faithfulness, the circuit must include all edges that result in the maximum gate effects, namely all edges from the AND, ADDER, and OR gates. However, considering sparsity, the gate effect sum remains maximal even if only one edge from each OR gate is retained.

### C.2 Completeness

Similarly, completeness requires that $D(G \setminus C||G)$ be maximized. Consider the following scenarios:

- For any gate $\{(A_1, A_2), B\}$, if the circuit does not include all edges or nodes from this gate, it is always possible to find a circuit $C^* = C \cup A_1, A_2, B$ such that $D(G \setminus C||G) < D(G \setminus C^*||G)$.

- For an AND gate $\{(A_1, A_2), B\}$, if the circuit $C$ only includes $A_1$ and $B$, then $G \setminus C$ will only contain $A_2$ or the edge $A_2 \rightarrow B$ (depending on whether pruning is applied to edges or nodes). Due to the AND operation, $B$ will not produce a gate effect. Therefore, for the circuit $C^* = C \cup A_2$, we have $D(G \setminus C||G) = D(G \setminus C^*||G)$.

- For an OR gate $\{(A_1, A_2), B\}$, if the circuit $C$ only includes $A_1$ and $B$, then $G \setminus C$ will only contain $A_2$ or the edge $A_2 \rightarrow B$. Due to the OR operation, $B$ still produces a gate effect. Therefore, it is always possible to find a circuit $C^* = C \cup A_2$ such that $D(G \setminus C||G) < D(G \setminus C^*||G)$.

- For an ADDER gate $\{(A_1, A_2), B\}$, if the circuit $C$ only includes $A_1$ and $B$, then $G \setminus C$ will only contain $A_2$ or the edge $A_2 \rightarrow B$. Due to the ADDER operation, $B$ still produces a gate effect. Therefore, it is always possible to find a circuit $C^* = C \cup A_2$ such that $D(G \setminus C||G) < D(G \setminus C^*||G)$.

Thus, to achieve optimal completeness, the circuit must include all edges that result in the maximum gate effects, namely all edges from the AND, ADDER, and OR gates. However, considering sparsity, the total gate effect remains maximized even if only one edge is retained for each AND gate.

# D    Validation of Logical Gates

## D.1    Toy Model

Motivated by [2], to study a toy transformer model with an AND, OR, and ADDER gates, we take a 1-Layer transformer model with two heads per layer, ReLU-based activations, and model dimension 1. Specifically, as shown in Figure 7, Let $A_1$ and $A_2$ be two attention heads with respective biases $bias_1$ and $bias_2$, both set to 1. The activation function $m$ is based on the ReLU nonlinearity. To ensure that the output of each attention head corresponds directly to its bias, we use a zero tensor as the input. For corrupted activations, we employ zero ablation—i.e., we directly remove the activations along the corresponding edges.

The activation function $m$ is configured differently to simulate logical gates as follows:

- **AND** gate: $m(x) = \text{ReLU}(x - 1)$. Under this setting, the output is 1 only when both $A_1$ and $A_2$ are active (i.e., not ablated); otherwise, the output is 0.

- **OR** gate: $m(x) = 1 - \text{ReLU}(1 - x)$. Here, the output is 1 as long as at least one of $A_1$ or $A_2$ is active; if both are ablated, the output is 0.

- **ADDER** gate: The $bias_2$ is modified to 1.5, and $m(x) = \text{ReLU}(x)$. In this case, the output is 0 when both $A_1$ and $A_2$ are ablated; it is 1.5 when only $A_1$ is ablated, 1 when only $A_2$ is ablated, and 2.5 when both are active.

Under these configurations, we evaluate the performance of existing methods on the toy model, as summarized in Table 1. For example, under the default Ns, ACDC [2] (representing greedy search), EAP [8] (representing linear estimation), and EdgePruning [10] (representing differentiable mask) all successfully identify both $A_1$ and $A_2$ in the AND and ADDER gates. However, in the OR gate, ACDC and EdgePruning identify only one of $A_1$ or $A_2$—the specific result depends on the search order in ACDC and the initialization of the mask in EdgePruning—while EAP fails to identify any high-effect edge.

Conversely, when these methods are executed under Dn, the outcomes are reversed. In the OR and ADDER gates, all three methods, ACDC, EAP and EdgePruning, can now identify both $A_1$ and $A_2$. However, in the AND gate, only ACDC and EdgePruning are able to recover one of $A_1$ or $A_2$, whereas EAP considers the effects of both to be insufficiently strong.

Ns can at least guarantee the full recovery of AND and ADDER gates. Greedy search, by retaining previous steps with removed results, can identify one OR

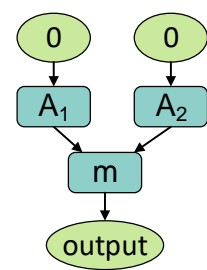

Figure 7: A toy model to study AND, OR, and ADDER gates.

edge (a detailed analysis is provided in Appendix B). Differentiable mask, optimizing for faithfulness (as per Corollary 1), ensures that at least one OR gate is included (otherwise, optimal faithfulness cannot be achieved), while additional OR edges conflict with the sparsity constraint and are therefore removed. However, linear estimation, when computing the effect of each edge, keeps all other edges in their non-removed state, which results in a failure to detect the OR gate. For example, when computing the effect of the edge $A \rightarrow C$ in the OR gate $A \rightarrow C \leftarrow B$, the edge $C \leftarrow B$ is also in the clean activation state, leading to a very small effect for $A \rightarrow C$. Similarly, when computing the effect of $C \leftarrow B$, the edge $A \rightarrow C$ remains in the clean activation state. In summary, linear estimation is unable to detect any edges of the OR gate.

The performance on Dn is the opposite of that on Ns, where, in addition to the full OR gates and ADDER gates, greedy search and differentiable mask can similarly recover one AND edge, just as they could with the OR gates in Ns. Linear estimation also fails to detect the AND gates due to the non-removed status of the other edges. Based on Corollary 1, we are also able to derive the performance of the three types of methods in terms of faithfulness and completeness.

## E    Experiment Details

### E.1    Why Does Search Strategy by repeating interventions on combinations is NP Problem?

Due to the differences in circuit algorithms, this issue cannot be fully proven mathematically. However, we will provide some additional explanation: we treat each ablation as $O(1)$. Assuming the language model contains $N$ nodes, if we need to use all possible intervention combinations to obtain a complete circuit, the time complexity would be $O(2^N)$, which far exceeds polynomial time. However, different strategies can optimize the solution verification process to varying degrees. For example, EAP only requires $O(N)$ time to verify whether the circuit is complete, while Edge-pruning involves gradient descent, making its time complexity difficult to estimate. Additionally, greedy search can only perform a greedy verification in $O(N)$ time rather than a thorough verification. Therefore, we ultimately classify this problem as NP problem.

However, our proposed method, which employs the algorithm $N_s + D_n$, does not introduce additional computational complexity to the existing circuit discovery algorithm. Conceptually, it is equivalent to reapplying the $N_s$ algorithm once more under the strategy of $D_n$. As a result, the overall time complexity increases by at most a factor of two relative to the baseline algorithm, without imposing any additional nonlinear burden.

### E.2    Detailed Results of Figure 3

Table 5: The changes in KL divergence between the original output and the new output when one or two edges are ablated.

| AND | | | OR | | | ADDER | | |
|---|---|---|---|---|---|---|---|---|
| edge1(state) | edge2(state) | ($\Delta$ KL)(state) | edge1(state) | edge2(state) | ($\Delta$ KL)(state) | edge1(state) | edge2(state) | ($\Delta$ KL)(state) |
| remain (1) | remain(1) | 0.50 (1) | remain (1) | remain(1) | 0.28 (1) | remain (1) | remain(1) | 0.65 (2) |
| remain (1) | ablate(0) | 0.02 (0) | remain (1) | ablate(0) | 0.27 (1) | remain (1) | ablate(0) | 0.32 (1) |
| ablate (0) | remain(1) | 0.04 (0) | ablate (0) | remain(1) | 0.27 (1) | ablate (0) | remain(1) | 0.36 (1) |
| ablate (0) | ablate(0) | 0.00 (0) | ablate (0) | ablate(0) | 0.00 (0) | ablate (0) | ablate(0) | 0.00 (0) |

All three subcircuits exhibit a fundamental pattern: the effect on the output is maximized when both edges are retained, and it drops to zero when both edges are ablated (as there is no clean activation left to support the node in this case). However, ablation of a single edge yields divergent outcomes:

In AND gate, ablating either edge nearly eliminates the effect on the output. In OR gate, ablating either edge has almost no impact on the output. In ADDER gate, ablating either edge leads to a noticeable degradation in the output.

### E.3    Baselines

In this work, we select **ACDC** [2] to represent greedy search methods, EAP [8] to represent linear estimation methods, and EdgePruning [10] to represent differentiable mask methods. In the following

sections, we provide a detailed exposition of the original design of each method under the Ns. strategy, the corresponding formulation under the Dn. strategy, and the final approach that integrates both—Ns.+Dn.—for recovering logically complete gates.

### E.3.1 Greedy Search Example: ACDC

The ACDC method identifies important edges by iteratively removing each edge and observing the effect of this intervention on the model output. Edges whose removal causes an effect greater than a predefined threshold $\tau$ are retained, while those with an effect smaller than $\tau$ are pruned. The original algorithm (Ns. strategy), is outlined as follows:

---

**Algorithm 1:** The ACDC algorithm in Ns.

---

**Data:** Computational graph $\mathcal{G}$, dataset $(x_i)_{i=1}^n$, corrupted datapoints $(x_i')_{i=1}^n$ and threshold
$\quad\quad \tau > 0$.
**Result:** Subgraph $\mathcal{H} \subseteq \mathcal{G}$.
1   $\mathcal{H} \leftarrow \mathcal{G}$          // Initialize H to the full computational graph
2   $\mathcal{H} \leftarrow \mathcal{H}.reverse\_topological\_sort()$        // Sort H so output first
3   **for** $v \in \mathcal{H}$ **do**
4      **for** $w$ *parent of* $v$ **do**
5         $\mathcal{H}_{\text{new}} \leftarrow \mathcal{H} \setminus \{w \to v\}$      // Temporarily remove candidate edge
6         **if** $D_{KL}(\mathcal{G}||\mathcal{H}_{\text{new}}) - D_{KL}(\mathcal{G}||\mathcal{H}) < \tau$ **then**
7            $\mathcal{H} \leftarrow \mathcal{H}_{\text{new}}$      // Edge is unimportant, remove permanently

8   **return** $\mathcal{H}$

---

In the **Ns.** strategy, $\mathcal{G}$ denotes the **clean run**, and $\mathcal{H} \setminus \{w \to v\}$ represents the replacement of the clean activation on the edge $w \to v$ with its corrupted activation. In contrast, under the **Dn.** strategy, $\mathcal{G}$ refers to the **corrupted run**, and $\mathcal{H} \setminus \{w \to v\}$ indicates the substitution of the corrupted activation on edge $w \to v$ with the corresponding clean activation.

In the combined **Ns.+Dn.** approach, the effects from both strategies are jointly considered. Specifically, the original pruning condition $D_{KL}(\mathcal{G} \,\|\, \mathcal{H}_{\text{new}}) - D_{KL}(\mathcal{G} \,\|\, \mathcal{H}) < \tau$ is replaced with the aggregated criterion: $D_{KL}(\mathcal{G}^{\text{clean}} \,\|\, \mathcal{H}_{\text{new}}) - D_{KL}(\mathcal{G}^{\text{clean}} \,\|\, \mathcal{H}) + D_{KL}(\mathcal{G}^{\text{corrupted}} \,\|\, \mathcal{H}_{\text{new}}) - D_{KL}(\mathcal{G}^{\text{corrupted}} \,\|\, \mathcal{H}) < \tau$.

### E.3.2 Linear Estimation Example: EAP

The EAP method approximates the effect of each edge using the first-order term of its Fourier expansion, enabling the estimation of all edge effects with a single forward pass. It is important to note that, during the computation of each edge's effect, all other edges remain in their unpruned (active) state.

Specifically, Ns. has approximation:

$$L(x|do(\tilde{x}_i)) - L(x) \approx (\tilde{x}_i - x_i)^T \frac{\partial}{\partial x_{i]}} L(x) \tag{4}$$

and Dn. has approximation:

$$L(\tilde{x}|do(x_i)) - L(\tilde{x}) \approx (\tilde{x}_i - x_i)^T \frac{\partial}{\partial \tilde{x}_{i]}} L(\tilde{x}) \tag{5}$$

Therefore, the approximation for Ns.+Dn. is $(\tilde{x}_i - x_i)^T \frac{\partial}{\partial x_{i]}} L(x) + (\tilde{x}_i - x_i)^T \frac{\partial}{\partial \tilde{x}_{i]}} L(\tilde{x})$.

### E.3.3 Differentiable Mask Example: EdgePruning

EdgePruning assigns a learnable mask to each node or edge, where the mask is reparameterized using the hard concrete distribution. In the Ns. setting, the optimization objective corresponds to Equation 1. Consequently, the objectives for the Dn. and Ns.+Dn. settings are given by Equation 2 and Equation 3, respectively.

In the Ns.+Dn. setting, directly optimizing both objectives jointly can lead to gradient interference and convergence to Pareto-optimal solutions, rather than a unified optimum. To address this, we independently compute the final mask values for Ns. and Dn. using Equations 1 and 2, and then obtain the mask for Ns.+Dn. by averaging the two.

# F    Misalignment Score

**Misalignment of AND**: For any subcircuit $\mathcal{K}_{\text{AND}} \subset \mathcal{C}_{\text{AND}}$, $\mathcal{C}_{\text{AND}}^* = \mathcal{C}_{\text{AND}} \setminus \mathcal{K}_{\text{AND}}$. Let $i, j \in \mathcal{C}_{\text{AND}}$, $i^*, j^* \in \mathcal{C}_{\text{AND}}^*$ be any two edges with the same receiver, respectively. The score of misalignment of AND reads:

$$\mathbb{E}_{i,j}[D(\mathcal{C}_{\text{AND}} \setminus i || \mathcal{C}_{\text{AND}} \setminus i, j)] - \mathbb{E}_{i^*,j^*}[D(\mathcal{C}_{\text{AND}}^* \setminus i^* || \mathcal{C}_{\text{AND}}^* \setminus i^*, j^*)] \tag{6}$$

Equation 6 indicates that the higher the score, the higher the misalignment. This is due to their properties: the effect caused by removing one and two edges from the AND gates is similar, while the effect for the ADDER gates differs significantly.

**Misalighment of OR**: Similarity, let any $\mathcal{K}_{\text{OR}} \subset \mathcal{C}_{\text{OR}}$, $\mathcal{C}_{\text{OR}}^* = \mathcal{C}_{\text{OR}} \setminus \mathcal{K}_{\text{OR}}$, $i, j \in \mathcal{C}_{\text{OR}}$, $i^*, j^* \in \mathcal{C}_{\text{OR}}^*$, respectively. The score of misalignment of OR reads:

$$\mathbb{E}_{i,i^*}[D(\mathcal{C}_{\text{OR}} \setminus i || \mathcal{C}_{\text{OR}}^* \setminus i^*)] - \mathbb{E}_{i,j,i^*,j^*}[D(\mathcal{C}_{\text{OR}} \setminus i, j || \mathcal{C}_{\text{OR}}^* \setminus i^*, j^*)] + m \tag{7}$$

Equation 7 utilizes the properties that the effect does not change by removing one edge from OR gates, while it significantly changes from ADDER gates. Additionally, to avoid the bias caused by both effects of ADDER and OR edges being marginally small in large-scale circuits, we replace the "difference in one edge" with the "difference in difference between one edge and two edges," and introduce a constant $m$ to ensure that the score $> 0$ (with $m$ set to 1.5 in practice).

Therefore, for any pair of $\mathcal{C}_{\text{Ns}}$ and $\mathcal{C}_{\text{Dn}}$, we can compute the misalignment using these two scores. We report the misalignment scores resulting from the intersection of Ns and Dn circuits at varying scales. Specifically, we select an Ns circuit consisting of 100 edges recovered from the IOI task and examine how the misalignment score changes as the number of edges in the Dn circuit varies from 60 to 140. Figure 8 illustrates that when Dn is significantly smaller than Ns, the misalignment score for the AND gates is high, as many ADDER edges are misclassified as AND edges. Conversely, when Dn is substantially larger than Ns, the misalignment score for the OR gates increases, due to many ADDER edges being misclassified as OR edges. When the number of edges in the Dn circuit is approximately equal to that in the Ns circuit, both the misalignment score and its standard deviation reach an acceptable level.

As shown in Figure 8, when the OR set significantly exceeds the AND set, a large number of misclassified ADDER gates appear only within the OR set, leading to a much higher misalignment score for OR compared to AND. The misalignment score is more of a "property," reflecting the optimal scale at which OR and AND sets should intersect. We determine the optimal ratio of AND to OR gates by minimizing the sum of the misalignment scores for AND and OR. The optimal results obtained with different methods and datasets are approximately as shown in Table 6:

Table 6: The optimal ratio of edge numbers between $\mathcal{C}_{\text{Ns}}$ and $\mathcal{C}_{\text{Dn}}$.

| strategies | IOI | GT | SA |
|---|---|---|---|
| ACDC | 1.12: 1 | 1.05: 1 | 1.07: 1 |
| EAP | 1.09: 1 | 1.03:1 | 1.05: 1 |
| Edge-Pruning | 1.11: 1 | 1.07: 1 | 1.06: 1 |

The results show that the proportion of AND is always slightly higher than that of OR, but it can be approximated as 1:1. Therefore, throughout this paper, we assume that the optimal alignment occurs when Ns and Dn contain an **equal number of edges**.

# G    Experiments of Completeness

In this Appendix, we present the detailed results of the completeness validation, together with a comparison against the Dn strategy. The specific results are reported in Tables 7 and 8. In brief,

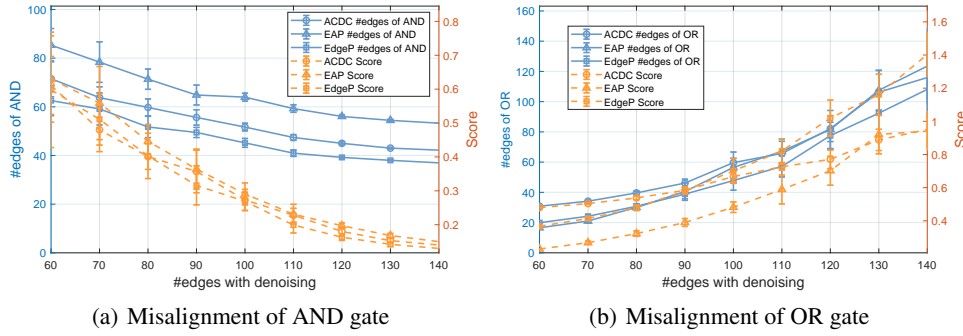

(a) Misalignment of AND gate          (b) Misalignment of OR gate

Figure 8: Misalignment score with 100-edges IOI circuit from Ns.

our method generally outperforms the approach based on Dn. Although the Dn-based method is theoretically capable of identifying all OR gates, it suffers from certain shortcomings in practice. For instance, as shown in Table 1, EAP fails to identify one of the edges associated with an AND gate under the Dn framework. Additionally, the greedy strategy may miss some OR gates located beneath branches of AND gates due to its dependence on the search order. The differentiable mask approach, on the other hand, suffers from the drawback that, in gates involving multiple edges, the mask values assigned to each edge tend to be lower, reducing its effectiveness.

Table 7: KL in Completeness Validation of IOI task, including Ns, Dn, and Ns+Dn.

| #edges | ACDC_NS | ACDC_Dn | ACDC_NsDn | EAP_Ns | EAP_Dn | EAP_NsDn | EdgeP_Ns | EdgeP_Dn | EdgeP_NsDn |
|--------|---------|---------|-----------|--------|--------|----------|----------|----------|------------|
| 100 | 0.62±0.1 | 0.64±0.1 | 0.64±0.1 | 0.51±0.1 | 0.71±0.1 | 0.74±0.1 | 0.56±0.1 | 0.62±0.1 | 0.64±0.1 |
| 200 | 0.64±0.1 | 0.69±0.1 | 0.71±0.1 | 0.56±0.1 | 0.73±0.0 | 0.73±0.1 | 0.58±0.1 | 0.65±0.1 | 0.69±0.1 |
| 500 | 0.71±0.1 | 0.84±0.1 | 0.87±0.1 | 0.65±0.1 | 0.81±0.1 | 0.83±0.1 | 0.67±0.1 | 0.88±0.1 | 0.94±0.1 |
| 1000 | 0.81±0.1 | 0.96±0.1 | 1.03±0.1 | 0.74±0.1 | 0.96±0.1 | 1.02±0.1 | 1.08±0.1 | 1.11±0.1 | 1.13±0.1 |
| 2000 | 1.05±0.1 | 1.19±0.2 | 1.22±0.1 | 0.89±0.1 | 1.11±0.2 | 1.24±0.2 | 1.31±0.2 | 1.40±0.1 | 1.41±0.1 |
| 5000 | 1.34±0.2 | 1.42±0.2 | 1.46±0.2 | 1.21±0.1 | 1.42±0.2 | 1.48±0.2 | 1.62±0.2 | 1.73±0.1 | 1.73±0.1 |

Table 8: Accuracy in Completeness Validation of IOI task, including Ns, Dn, and Ns+Dn.

| #edges | ACDC_NS | ACDC_Dn | ACDC_NsDn | EAP_Ns | EAP_Dn | EAP_NsDn | EdgeP_Ns | EdgeP_Dn | EdgeP_NsDn |
|--------|---------|---------|-----------|--------|--------|----------|----------|----------|------------|
| 100 | 0.73±0.1 | 0.68±0.0 | 0.63±0.1 | 0.74±0.1 | 0.53±0.0 | 0.51±0.0 | 0.76±0.1 | 0.67±0.1 | 0.66±0.0 |
| 200 | 0.65±0.0 | 0.62±0.0 | 0.60±0.0 | 0.68±0.0 | 0.49±0.0 | 0.46±0.0 | 0.59±0.0 | 0.55±0.0 | 0.56±0.0 |
| 500 | 0.53±0.0 | 0.51±0.0 | 0.51±0.0 | 0.61±0.0 | 0.41±0.0 | 0.38±0.0 | 0.55±0.0 | 0.43±0.0 | 0.43±0.0 |
| 1000 | 0.40±0.0 | 0.34±0.0 | 0.33±0.0 | 0.39±0.0 | 0.29±0.0 | 0.26±0.0 | 0.32±0.0 | 0.27±0.0 | 0.26±0.0 |
| 2000 | 0.34±0.0 | 0.26±0.0 | 0.24±0.0 | 0.35±0.0 | 0.18±0.0 | 0.14±0.0 | 0.24±0.0 | 0.18±0.0 | 0.16±0.0 |
| 5000 | 0.26±0.0 | 0.22±0.0 | 0.21±0.0 | 0.29±0.0 | 0.13±0.0 | 0.09±0.0 | 0.21±0.0 | 0.11±0.0 | 0.11±0.0 |

# H  Experiments of Faithfulness

In this section, we investigate the faithfulness of circuits obtained using three methods—ACDC, EAP, and EdgePruning—across three tasks: IOI, GT, and SA. Specifically, we examine the changes in KL divergence and accuracy between the original circuit (Ns.), the circuit with full OR and ADDER gates (Dn.) and the circuit with logically complete gates (Ns.+Dn.). For sparsity, we select edge counts of 100, 200, 500, 1000, 2000, and 5000.

Figure 9 illustrates that, under the same sparsity constraints, the circuits discovered using Dn. are significantly lower in both metrics compared to those discovered using Ns. and Ns.+Dn., which corroborates our assertion in Corollary 1: Dn. is incapable of fully recovering the AND gate, and thus cannot achieve optimal faithfulness.

Additionally, in the EAP method, Ns clearly performs much worse than Ns+Dn, whereas in the ACDC and EdgePruning methods, the performance of Ns and Ns+Dn is quite similar. This aligns with our reasoning in Table 1, where we note that only the linear estimation method completely fails to identify any OR edge, thus not satisfying the minimal requirement of faithfulness.

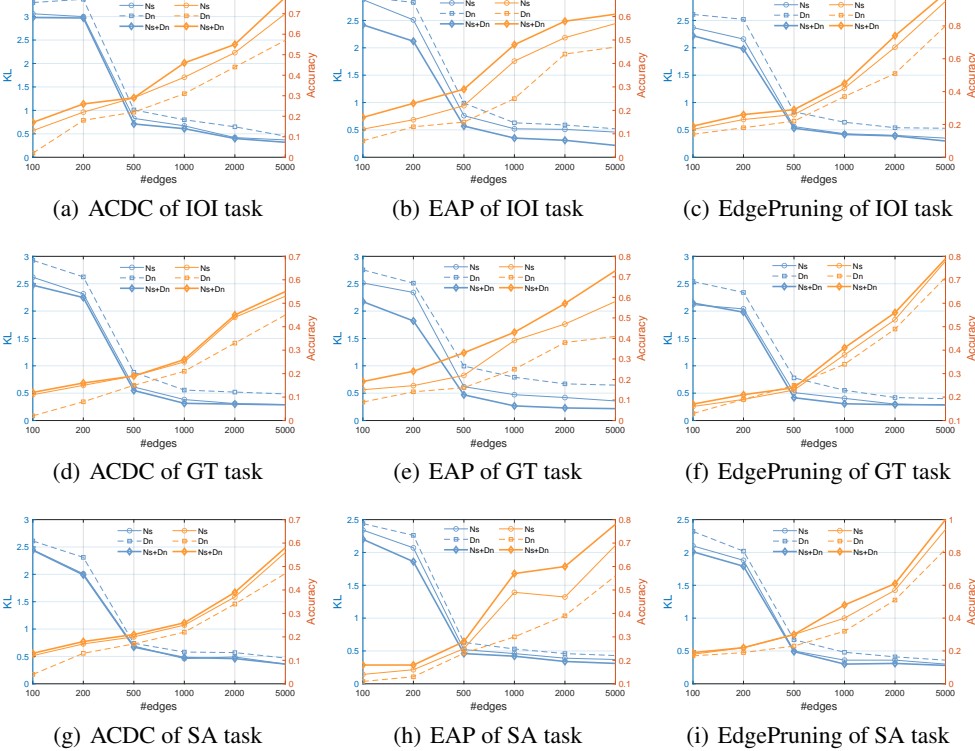

Figure 9: Faithfulness of circuit from Ns., DN., and NS.+Dn..

# I  Graph Study

To investigate the relationship between the three logical gates and the functions of the language model, we extracted the circuits of AND, OR, and ADDER gates discovered through ACDC on the IOI task. We then reviewed the IOI circuit [16] to determine the function of each component.

Interestingly, each receiver in the AND circuit is almost always influenced by edges from **different** functions, indicating that the AND operation can be understood as combining different functions to jointly impact the subsequent layers. For example, in Figure 10(a), the Induction Head requires edges from both the Duplication Token Head and the Previous Token Head to function, which supports the mechanism behind the Induction skill [27, 28, 29]. Similarly, the Name Mover Head requires support from both the S-Inhibition Head and the Induction Head, which explains the functional mechanism of the AND operation.

In contrast, the OR circuit clearly shows that nearly every receiver node is influenced by edges from the **same** function, as shown in in Figure 10(b), suggesting that these edges from the same function are either backups or interchangeable. For instance, the S-Inhibition Head is influenced by multiple Induction Heads, and the Backup Name Mover Head is influenced by multiple S-Inhibition Heads.

Lastly, the ADDER circuit appears to focus more on the outputs of the **MLP** and often combines outputs from shallow-layer skills with those from deeper-layer skills, as shown in Figure 10(c). The Name Mover Head considers outputs from all functions between the Duplicate Token Head and the S-Inhibition Head, and the final output takes into account the combined results from all three Name Mover Heads.

Additionally, regarding the span of gates across layers, the OR gates typically operate over the **shortest** distances, usually occurring between two functions that are close in layer position. In contrast, the ADDER gates generally span the **longest** distances, typically combining shallow-layer functions with deeper-layer functions.

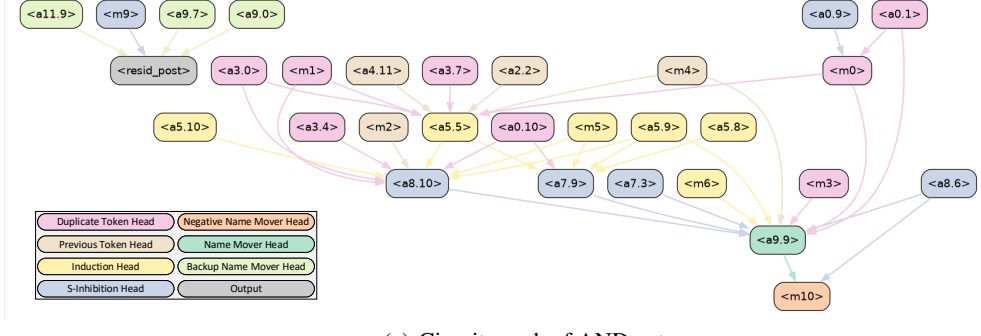

(a) Circuit graph of AND gate

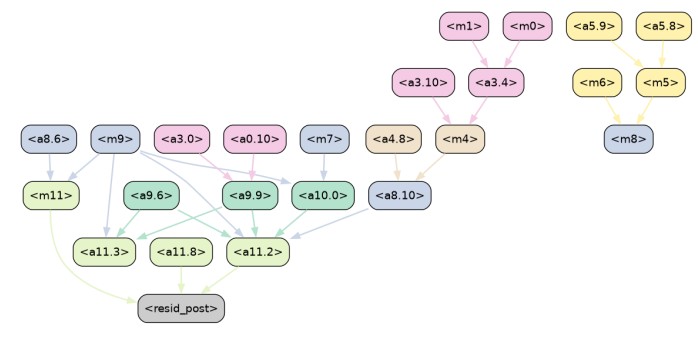

(b) Circuit graph of OR gate

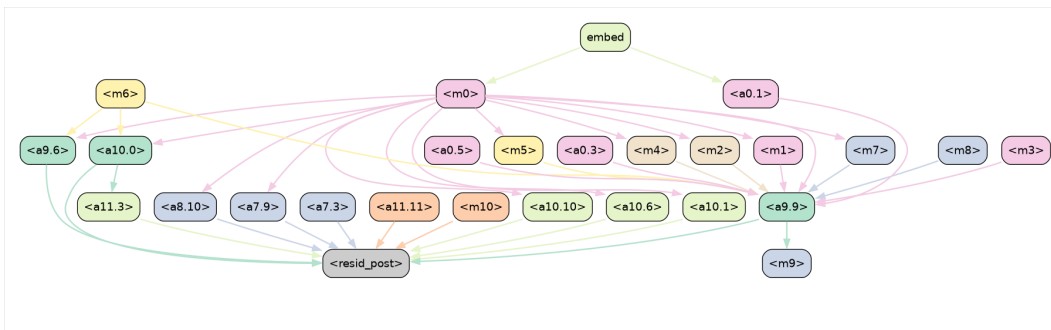

(c) Circuit graph of ADDER gate

Figure 10: Circuit Graphs of AND, OR, and ADDER gates, respectively. We set the color of each component to be the same as that of the IOI circuit [16], allowing for easy reference to the function of each component.

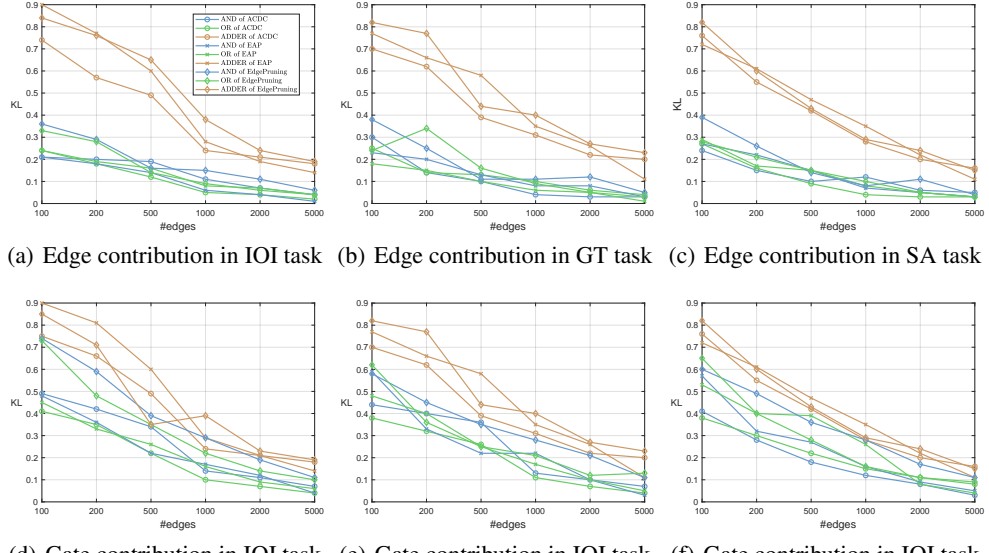

| (a) Edge contribution in IOI task | (b) Edge contribution in GT task | (c) Edge contribution in SA task |
| (d) Gate contribution in IOI task | (e) Gate contribution in IOI task | (f) Gate contribution in IOI task |

Figure 11: The contribution of different logic gates to the output."

## J Output Contribution

In this section, we investigate the contribution of three types of logic gates to the output. Specifically, we calculate the change in the KL divergence of the output caused by replacing these gates. Considering the collection of gates, we analyze their contributions from two perspectives: the **gate effect** and the **edge effect**. The gate effect refers to the impact on the output caused by replacing an entire logic gate, while the edge effect corresponds to equally distributing the gate effect across each edge within the gate. For example, for an AND gate with two edges, if the activation of the receiver node contributes 0.8 to the output, the edge effect would be 0.4. Figure 11 illustrates the gate effect and edge effect of these logic gates across different tasks and baselines. Clearly, the ADDER gates exhibit the largest contribution, demonstrating its role as the primary framework of the circuit, while the contributions of the AND and OR gates are similar. Additionally, the average gate and edge effects in EdgePruning and EAP are significantly higher than those in ACDC. This is because the differentiable mask and linear estimation methods optimize (rank) based on the edge effect, ensuring that the effect within the circuit is maximized, in contrast to greedy search methods.

## K Proportion of AND, OR, and ADDER Gates

Figure 12 illustrates the proportion of the three types of logical gates across different tasks for each method. Notably, in the ACDC baseline, the number of edges corresponding to each gate type is nearly equal. This is because ACDC employs a greedy search strategy without ranking edges by their effect on the output. In contrast, both EAP and EdgePruning yield significantly fewer OR edges, reflecting the fact that OR edges contribute the least to the output—a finding we detail in Section 5.2 and Appendix J. Furthermore, the results from EdgePruning indicate that the number of AND edges is similarly low, comparable to OR edges. This arises from the fact that EdgePruning optimizes based on individual edge effects rather than gate-level effects. For instance, in a gate comprising two AND edges, each contributes only half of the total gate effect. As a result, during optimization, the mask values for such edges may be suppressed, increasing the likelihood of pruning.

## L Potential Contributions of Completeness: an Example of Model Unlearning

We have designed a toy task analogous to knowledge unlearning to demonstrate its actual effectiveness. Specifically, given a harmful-response datasets (PKU-SafeRLHF) [30], we first obtain its circuit $\mathcal{C}$, and then remove this circuit from the computation graph $\mathcal{G}$, observing whether the model

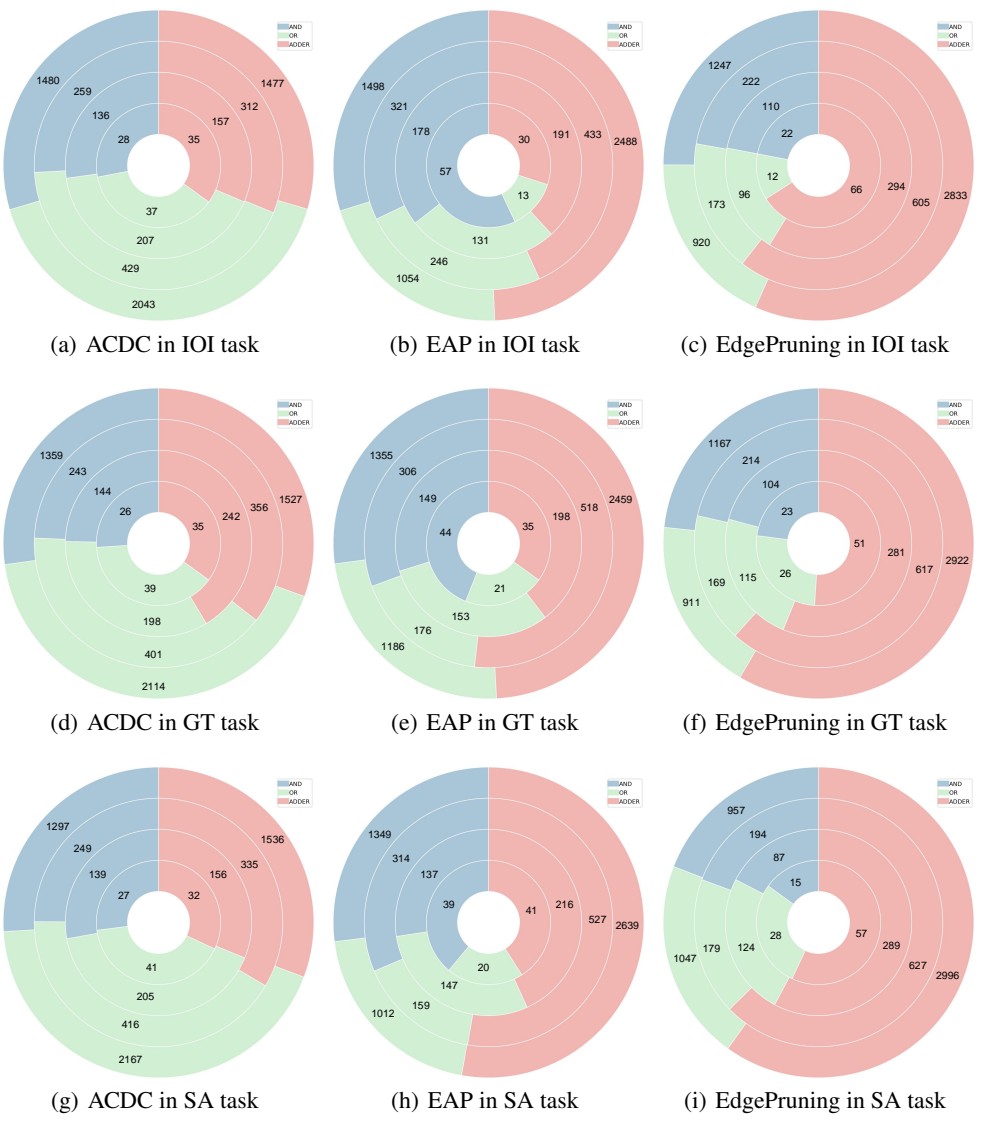

Figure 12: Proportion of AND, OR, and ADDER edges in circuit from Ns.+Dn., The concentric rings from the innermost to the outermost represent circuits with 100, 500, 1000, and 5000 edges, respectively. The blue represents AND edges, red represents ADDER edges, and green represents OR edges.

can still respond to the dataset. The dataset provides harmful responses for certain questions, which are used to evaluate the efficacy of preventing the model from generating these harmful responses.

In the Table 9, we present the performance of this toy task on the accuracy of $\mathcal{G} - \mathcal{C}$, which refers to the performance of the computational graph after the circuit has been removed (using the interchange ablation method), reflecting the completeness of the circuit. We test two circuit discovery methods: EAP and Edge-Pruning.

Table 9: A toy experiment on model unlearning task

| Method | accuracy | accuracy of G-C |
|--------|----------|-----------------|
| EAP | 92.74 | 29.17 |
| Edge-P | 92.74 | 31.44 |
| Ours | 92.74 | 13.21 |

It is evident that, compared to existing circuit discovery methods, our method ensures that the computational graph loses almost all the necessary mechanisms for the task once the circuit is removed. Additionally, this toy task highlights the significance of studying circuit completeness, which holds great potential in unlearning harmful information: a complete circuit can help us pinpoint all neurons associated with harmful responses.

