# OpenReview forum: "Rethinking Circuit Completeness in Language Models: AND, OR, and ADDER Gates"
_NeurIPS.cc/2025/Conference — NeurIPS 2025 poster_

### Official Review · Reviewer_ftP7 · 2025-07-03

**Clarity:** 2
**Significance:** 2
**Originality:** 3
**Rating:** 4
**Confidence:** 3

**Summary:**

This paper proposes a new way to interpret circuits in terms of three types of logical gate, which help shed light on and tease apart the important concepts of completeness and faithfulness. They explain how these gate types relate to the circuits found by existing methods, and propose a new method that compromises between existing methods to efficiently find better circuits by leveraging this new understanding. They apply these ideas to a number of well-studied tasks in mechanistic interpretability to validate key claims and show that these new concepts and methods can lead to new insights and improved interpretability in practice.

**Questions:**

See weaknesses above, particularly the last one.

**Ethical Concerns:**

["NO or VERY MINOR ethics concerns only"]

**Final Justification:**

I appreciate the authors' explanations and now feel I understand the paper's contribution much better. I think the authors present an interesting idea, but now that I understand these gates better I'm a bit less convinced of their interpretability impact, so I will raise my score only to a 4.

**Limitations:**

Yes.

**Quality:**

3

**Strengths And Weaknesses:**

Strengths:
1. I found section 2 in particular to be a well-written, clear, and concise summary of background on terms and methods for this paper.
2. The work is well-motivated—I agree it is important to have a better understanding of completeness and faithfulness, and more generally interpretability, of discovered circuits. And furthermore this seems to be a novel contribution.
3. The empirical demonstrations seem to be compelling, particularly Fig 3 and the conclusions drawn in 5.1, in showing that these classes of gates exist, can be identified, and can lead to meaningful insights that wouldn’t otherwise be achievable.

Weaknesses:
1. Legend in Fig 4 is unreadably small
2. I don’t understand lines 144-148: what does it mean “B can influence the result”? And why should an OR gate remain functional as long as at least one edge remains intact? The function f(A_1, A_2) = A_1 ^ A_2 is not the same as the function g(A_1,A_2) = A_1, so I don’t see why it should be the case that an OR gate doesn’t care if one edge is dropped. Actually, it’s not even clear to me what function is being used when, say, A_2 is “dropped”—I guess it depends on the ablation method, but surely it’s not true that no matter what ablation method is used, ablating one edge in an OR gate doesn’t change it at all?
3. There is a claim (repeated twice) that the simplest approach to uncovering OR gates would make circuit discovery an NP problem, but no reference or explanation is given.
4. Line 153 states that any circuit CAN be represented via the proposed gates, but I don’t see how that means that every circuit IS represented via the proposed gates in terms of the nodes and edges it is actually defined by. Like, in order to represent a given circuit via only these gates, it seems you’d have to make a new graph that is computationally equivalent to the old graph but only uses these three gates; it’s not clear how to do this, nor that one would want to do this, since nodes and edges in the new graph will no longer have the same interpretation as those in the old graph.
5. I think I have a high-level, very fundamental point of confusion about this paper. It seems to use the language of binary computation throughout (e.g., words like “logic” and even the names of the gates “and”, “or”), yet as far as I know this is not what the circuits people discover look like, i.e., the information being passed along edges is not binary, but instead is continuous, and in general multidimensional. I think this must be the case in their experiments as well, since they use GPT2-small, and Fig 6’s nodes are entire attention heads or MLPs. So then is the binary/logical language just sort of figurative/suggestive for classifying the components of these circuits and interpreting them? This disconnect between binary intuition/motivation and the reality of non-binary components in actual circuits seems to be unaddressed in the paper, which I found surprising given that it seems to otherwise be clearly and thoughtfully written—I feel I am missing something simple but major, which may also account for a number of weaknesses I listed above. So if the authors can clarify this, it could significantly impact my score.

---

> ### Author Rebuttal · Authors · 2025-07-28
>
> Thank you for your review. From your feedback, we understand that your **primary concern** lies in how the circuit gates model "binary-like" mathematical structures from continuous values (Q2, Q5). Your questions have provided valuable insights for improving the clarity of our paper, for which we are sincerely grateful. Below, we will first focus on addressing the modeling of circuit gates, followed by responses to the other issues you raised.
>
> ### **Q1: This disconnect between binary intuition/motivation and the reality of non-binary components in actual circuits seems to be unaddressed in the paper. (weakness 2 and 5 in review text)**
>
> Thank you for your suggestion! We will reinterpret the logical gates from the perspectives of Ns (Nosing) and Dn (Denoising) for you. Specifically, for any given dataset, we can derive two circuits: $C_{Ns}$ and $C_{Dn}$, using Ns and Dn. By examining the intersection of $C_{Ns}$ and $C_{Dn}$, we obtain three subcircuits:
>
> - **Subcircuit 1**: edges that are present in $C_{Ns}$ but absent from $C_{Dn}$.
> - **Subcircuit 2**: edges that are present in $C_{Dn}$ but absent from $C_{Ns}$.
> - **Subcircuit 3**: edges that are shared between $C_{Ns}$ and $C_{Dn}$.
>
> For these three subcircuits, we separately calculated the changes in KL divergence between the original output and the new output when one or two edges are ablated. The following three tables present the mean values of the samples obtained using the EAP method on the IOI dataset. For each subcircuit, we randomly selected 30 "three-node" ($A1, A2, B$) samples, with the structure $A1 \rightarrow B \leftarrow A2$.
>
> **subcircuit 1** (edge1=$ A1 \rightarrow B$, edge2=$ A2\rightarrow B$):
> | edge1 | edge2   |($\Delta KL$)|
> | ------------- | --------------|----------------------------  |
> | remain    |  remain    | 0.50       |
> | remain     |  ablate    | 0.02       |
> | ablate     |  remain    | 0.04       |
> | ablate     |  ablate    | 0.00       |
>
> **subcircuit 2**:
> | edge1 | edge2   |($\Delta KL$)|
> | ------------- | --------------|----------------------------  |
> | remain     |  remain   | 0.28       |
> | remain     |  ablate    | 0.27      |
> | ablate     |  remain    | 0.27       |
> | ablate     |  ablate    | 0.00      |
>
> **subcircuit 3**
> | edge1 | edge2   |($\Delta KL$)|
> | ------------- | --------------|----------------------------  |
> | remain     |  remain    | 0.65       |
> | remain     |  ablate    | 0.32       |
> | ablate     |  remain    | 0.36       |
> | ablate     |  ablate   | 0.00      |
>
> All three subcircuits exhibit a fundamental pattern: the effect on the output is maximized when both edges are retained, and it drops to zero when both edges are ablated (as there is no clean activation left to support the node in this case). However, ablation of a single edge yields divergent outcomes:
>
> - In **subcircuit 1**, ablating either edge **nearly eliminates** the effect on the output.
> - In **subcircuit 2**, ablating either edge has **almost no impact** on the output.
> - In **subcircuit 3**, ablating either edge leads to **a noticeable degradation** in the output.
>
> To formalize this, we define a **state variable** where (see **(state)** in three tables):
> - **state = 1** denotes that an edge or node is **active**/**retained**/**exerts an effect**,
> - **state = 0** indicates that it is **inactive**/**ablated**/**has no effect**.
>
> Given that **subcircuit 3** exhibits a graded influence on the outcome, we further introduce:
> - **state = 2** to represent a significantly greater effect than **state = 1**.
>
> With these definitions, the combinations of states in the three subcircuits correspond respectively to the **truth tables** of an **AND gate**, an **OR gate**, and an **ADDER gate**, as the following tables.
>
> **subcircuit 1 (AND gate)**:
> | edge1 state | edge2 state   |receiver state |
> | ------------- | --------------|----------------------------  |
> | **1** (remain)    |  **1** (remain)    | **1** ($\Delta KL$=0.50)    |
> | **1** (remain)    |  **0** (ablate)    | **0** ($\Delta KL$=0.02)     |
> | **0** (ablate)     |  **1** (remain)    | **0** ($\Delta KL$=0.04)    |
> | **0** (ablate)     |  **0** (ablate)     | **0** ($\Delta KL$=0.00)    |
>
> **subcircuit 2 (OR gate)**:
> | edge1 state | edge2 state   |receiver state |
> | ------------- | --------------|----------------------------  |
> | **1** (remain)    |  **1** (remain)    |**1** ($\Delta KL$=0.28)       |
> | **1** (remain)    |  **0** (ablate)    | **1** ($\Delta KL$=0.27)        |
> | **0** (ablate)    |  **1** (remain)    | **1** ($\Delta KL$=0.27)         |
> | **0** (ablate)   |  **0** (ablate)   | **0** ($\Delta KL$=0.00)      |
>
> **subcircuit 3 (ADDER gate)**
> | edge1 state | edge2 state   |receiver state |
> | ------------- | --------------|----------------------------  |
> | **1** (remain)    |  **1** (remain)    | **2** ($\Delta KL$=0.65)      |
> | **1** (remain)    |  **0** (ablate)   | **1** ($\Delta KL$=0.32)       |
> | **0** (ablate)   |  **1** (remain)    | **1** ($\Delta KL$=0.36)       |
> | **0** (ablate)   |  **0** (ablate)   | **0** ($\Delta KL$=0.00)      |
>
> Accordingly, we refer to **subcircuit 1 as the AND gate**, **subcircuit 2 as the OR gate**, and **subcircuit 3 as the ADDER gate**. In existing works [1][2][3], **similar definitions** of the OR gate have been proposed, providing a **solid theoretical foundation** for our work. In contrast to these studies, **our contribution** lies in the fact that, in addition to the OR gate, we also introduce the AND and ADDER gates, thereby endowing **all nodes and edges** in the Ns and Dn circuits with **logical properties**.
>
> Moreover, this phenomenon is **not incidental**; it is consistently observed across all implementations of both Ns and Dn. In Appendices A and B, we prove that the emergence of AND, OR, and ADDER gates arises from mutually exclusive properties of Ns and Dn, and that they represent an **intrinsic logical structure** of the circuit graph itself.
>
> Finally, as shown in the table above, this also explains why the OR gate has almost no change on the effect when one of its edges is removed (when edge1=1,edge2=0, or conversely edge1=0,edge2=1).
>
> **reference:**
>
> *[1]. Conmy, Arthur, et al. "Towards automated circuit discovery for mechanistic interpretability." NeurIPS 2023.*
>
> *[2]. Wang, Kevin Ro, et al. "Interpretability in the Wild: a Circuit for Indirect Object Identification in GPT-2 Small." ICLR 2023.*
>
> *[3]. Heimersheim, Stefan, et al. "How to use and interpret activation patching." arXiv 2024.*
>
> ### **Q2: Legend in Fig 4 is unreadably small**
>
> Thank you for your suggestion. We will include a table with specific quantitative numbers in revised versions to avoid making the figures too complex.
>
> ### **Q3: There is a claim (repeated twice) that the simplest approach to uncovering OR gates would make circuit discovery an NP problem, but no reference or explanation is given.**
>
> Thank you for your suggestion. Due to the differences in circuit algorithms, this issue cannot be fully proven mathematically. However, we will provide some additional explanation: we treat **each ablation** as $O(1)$. Assuming the language model contains $N$ nodes, if we need to use all possible intervention combinations to obtain a complete circuit, the time complexity would be $O(2^N)$, which far exceeds polynomial time. However, **different strategies can optimize the solution verification process to varying degrees**. For example, EAP only requires $O(N)$ time to verify whether the circuit is complete, while Edge-pruning involves gradient descent, making its time complexity difficult to estimate. Additionally, greedy search can only perform a greedy verification in $O(N)$ time rather than a thorough verification. Therefore, we ultimately classify this problem as NP problem.
>
> **In the revised version, we will incorporate these analyses into the main text to provide a simple explanation for the "NP problem."**
>
> ### **Q4: Line 153 states that any circuit CAN be represented via the proposed gates, but I don’t see how that means that every circuit IS represented via the proposed gates in terms of the nodes and edges it is actually defined by. Like, in order to represent a given circuit via only these gates, it seems you’d have to make a new graph that is computationally equivalent to the old graph but only uses these three gates; it’s not clear how to do this, nor that one would want to do this, since nodes and edges in the new graph will no longer have the same interpretation as those in the old graph.**
>
> Thank you for your question. In fact, representing a circuit using three types of gates does **not require making a new circuit**. We simply need to analyze the relationships between the edges in the current circuit, classifying them as one of the AND, OR, or ADDER gates based on their effect on the receiver node. As the simplest case, as shown in the tables in Q1, we only need to compute the impact of these edges on the receiver node in order to determine whether it belongs to the AND, OR, or ADDER gate.
>
> Therefore, the circuit with three types of gates is **structurally identical to the original circuit**. The reason we construct a new circuit using Ns+Dn is that the existing circuit is incomplete (due to the nature of Ns, which misses some branches of the OR gate). Moreover, as shown in the circuit graph in **Appendix G**, in the IOI task, we confirmed that the circuits completed using our method **contain all the nodes and edges originally present in the IOI circuit**. The additional components are almost entirely edges corresponding to "OR" gates that were not identified by the previous algorithm. As a result, the meaning and functionality of each edge and node remain consistent with the original IOI circuit, still corresponding to functions such as the duplicate head, induction head, and S-inhibition head.

---

> > ### Comment · Reviewer_ftP7 · 2025-08-02
> >
> > I thank the authors for their detailed response and for helping me better understand their paper. I think I now understand my key point of confusion: the logic gates described in the paper are not logic gates as a function of the information in the circuit, but are (approximately) logic gates as a function of the binary variables of whether or not each incoming edge is ablated (=0) or not (=1). My understanding of the authors' response to Q1 was a much clearer explanation of this, but I saw no plan or commitment to change the presentation of the paper to make this clearer. I'm not sure I can raise my score much without such a commitment/plan, as I find the current presentation quite misleading on this point, and it seems to me that other reviewers did too given their similar questions. As an example, in definition 1, you seem to write each node in a circuit as a function of its inputs ("sender nodes") A's, namely, the values of its parent nodes, yet you write the relationship between those inputs and the output B via logical notation and say things like "all sender nodes satisfy an OR logical relationship with the receiver node" suggesting the logical relationship is between the nodes in the circuit, which is not correct according to my revised understanding (but correct me if I'm wrong). In particular, each logical gate type does not uniquely identify a node's function of its parent nodes, nor does it even describe that function as being a logical function of its inputs, it only identifies a *property* of how that function's output impacts the final output when its inputs are ablated. To be clear, I think these properties are still interesting, but they're very different from my initial understanding, and when I go back and read the paper again with my revised understanding, I cannot see how a reader could have reached my revised understanding from the presentation.
> >
> > Now that (I think) I understand better what is meant by the logical gates in this paper, I remain very confused by the quote "any circuit can be represented as a combination of AND, OR, and ADDER gates", which seems to be a central claim in the paper. Other reviewers seem to wonder this too, but I'm not sure I understood the responses to them (Q4 of reviewer mnpW and Q1 of reviewer khCH). The authors' only justification for this quote seems to be that "The union of these three gates encompasses all the nodes and edges present in both Ns and Dn, and therefore, we may reasonably assume that they are complete." but I don't follow this argument. The authors' response to Q4 in their rebuttal to me seems to reinforce my interpretation of the quote from their paper as claiming that ANY circuit (i.e., computational graph), no matter how it is represented (e.g., nodes could represent single neurons and edges scalar-valued functions, or nodes could represent much more complex objects such as entire attention heads so that edges much be functions with higher-dimensional output spaces), can have EVERY node in it categorized as either an OR, AND, or ADDER gate. I find this hard to believe, and certainly if it is claimed to be true, it would require a thorough explanation/justification/proof in the paper, which is currently lacking.

---

> ### Author Response · Authors · 2025-08-03
> **Response**
>
> Thank you for your response! Your reply primarily concerns **two issues**.
> - **Q1**: You recommend that we clarify that the term **logical relationship merely identifies a property of how a function’s output impacts the final output when its inputs are ablated, rather than describes that function as being a logical function of its inputs**, in order to avoid potential misunderstandings for the reader.
> - **Q2**: You express concern regarding whether our logical gates continue to hold when applied to **circuit graphs at different levels of granularity**. We address both of these issues in detail below.
>
> ### Q1
>
> **We agree with your perspective** and apologize for the misunderstanding we caused. As you noted, **logical relationships and functional relationships are entirely distinct concepts**.
>
> We also **appreciate your suggestion that we provide a clear plan and commitment for revising the definition of circuit**. Accordingly, we have decided to revise lines 131–152 of the manuscript to better reflect our response to your concerns. Below, we **present the revised versions** of the circuit definition.
>
> **Revised version:**
>
> ***Definition 1**. We assumed a common paradigm that there exists a receiver node $B$, which is connected by more than 1 sender node $A_1$, $A_2$, . . . ,. For any edge $A_i → B$, we use binary values “0” and “1” to represent the activation state of a node. Specifically, $A_i=0$ indicates that node $A_i$ is removed, ablated, or deactivated, whereas $A_i=1$ indicates that node $A_i$ is retained and active.  When the sender nodes are ablated, the effect of node $B$ on the output exhibits **three** distinct patterns, which are as follows:*
>
> - ***AND gate**: All sender nodes satisfy an AND logical relationship with the receiver node, i.e., $B = A_1 ∧ A_2 ∧. . . $. In this case, node $B$ exerts a significant effect on the output only if all of its sender nodes are retained. If even a single sender node is ablated, the effect of B on the output is nearly eliminated. An example of its corresponding truth table is shown in Table 1(a).*
>
> - ***OR gate**: All sender nodes satisfy an OR logical relationship with the receiver node, i.e., $B = A_1 ∨ A_2 ∨. . . $. In this case, node $B$ always exerts a significant effect on the output if one or more of its sender nodes are retained. Only if all sender nodes are ablated, the effect of B on the output is nearly eliminated. An example of its corresponding truth table is shown in Table 1(b).*
>
> - ***ADDER gate**: all sender nodes satisfy an ADDER logical relationship with the receiver node, i.e., $B = A_1 + A_2 + . . .$. In this case, node $B$ exhibits its maximal effect on the output only when all of its sender nodes are retained. If any single sender node is ablated, the effect of $B$ on the output is substantially diminished; when all sender nodes are ablated, $B$'s effect on the output is reduced to zero. Accordingly, we define the state of $B$ as taking values 0,1,2,…, where the total number of distinct states equals the number of sender nodes. An example of its corresponding truth table is shown in Table 1(b).*
>
> *Table 1: the changes in KL divergence between the original output and the new output when one or two edges are ablated. Samples obtained using the EAP method on the IOI dataset. For each subcircuit, we randomly selected 30 "three-node" ($A1, A2, B$) samples, with the structure $A1 \rightarrow B \leftarrow A2$.*
>
> (a). AND gate
>
> | edge1(state) | edge2(state)   |($\Delta KL$)(state)|
> | ------------- | --------------|----------------------------  |
> | remain (1)    |  remain(1)    | 0.50 (1)      |
> | remain (1)    |  ablate(0)    | 0.02 (0)      |
> | ablate (0)    |  remain(1)    | 0.04 (0)      |
> | ablate (0)    |  ablate(0)    | 0.00 (0)      |
>
> (b). OR gate
>
> | edge1(state) | edge2(state)   |($\Delta KL$)(state)|
> | ------------- | --------------|----------------------------  |
> | remain (1)    |  remain(1)    | 0.28 (1)      |
> | remain (1)    |  ablate(0)    | 0.27 (1)      |
> | ablate (0)    |  remain(1)    | 0.27 (1)      |
> | ablate (0)    |  ablate(0)    | 0.00 (0)      |
>
> (c). ADDER gate
>
> | edge1(state) | edge2(state)   |($\Delta KL$)(state)|
> | ------------- | --------------|----------------------------  |
> | remain (1)    |  remain(1)    | 0.65 (2)      |
> | remain (1)    |  ablate(0)    | 0.32 (1)      |
> | ablate (0)    |  remain(1)    | 0.36 (1)      |
> | ablate (0)    |  ablate(0)    | 0.00 (0)      |
>
> *The circuit gates reveal the dependency structures underlying node existence. Notably, these gates are independent of the specific functions of the nodes. For example, in Figure 10(b), the nodes together form many OR gate, yet  their function can be induction, or duplication, or name moving. OR gates indicate that the circuit maintains approximately effect as long as at least one of the incoming edges is not ablated.*
>
> **We hope that the above revisions eliminate any potential misunderstandings from the original version.**

---

> > ### Comment · Reviewer_ftP7 · 2025-08-03
> >
> > I thank the reviewers for their response--they have correctly understood my questions and the revised text they committed to is an improvement. However, what I still find confusing is that there is no distinction in the language made between the node of the circuit (i.e., the actual subgraph of the computational model that performs the function of the model) and the binary value indicating its activation. For instance, Definition 1 begins with
> > "We assumed a common paradigm that there exists a receiver node $B$, which is connected by more than 1 sender node $A_1$, $A_2$, . . . ,. For any edge $A_i → B$, we use binary values “0” and “1” to represent the activation state of a node. Specifically, $A_i=0$ indicates that node $A_i$ is removed, ablated, or deactivated"
> > The first sentence defines the $A_i$'s and $B$ as nodes and the beginning of the second sentence alludes to edges between them. But the end of the second sentence and the third sentence define the value of a node $A_i$ as a binary value based on whether "node $A_i$ is removed, ablated or deactivated". So it seems to me that the latter two instances of the word "node" in the quotation above refer to the actual nodes in the circuit, whereas the word "node" in the first sentence refers to the binary variables $A_i$ and $B$. These binary variables are each associated to a node in the circuit, but are not themselves nodes in the circuit (the nodes in the circuit are not binary-valued).

---

> > > ### Author Response · Authors · 2025-08-03
> > > **Response**
> > >
> > > We sincerely appreciate your meticulous feedback. We understand that you are emphasizing the importance of distinguishing between **“the activation value of node $A$”** and **“the activation state of node $A$”** in our definition. To address this point, we have made a minor revision to Definition 1:
> > >
> > > **Original version**:
> > > “We assumed a common paradigm that there exists a receiver node $B$, which is connected by more than 1 sender node $A_1, A_2, \dots$. For any edge $A_i \rightarrow B$, we use binary values ‘0’ and ‘1’ to represent the activation state of a node. Specifically, $A_i = 0$ indicates that node $A_i$ is removed, ablated, or deactivated, ...”
> > >
> > > **Revised version**:
> > > “We assume a common paradigm in which a receiver node $B$ is connected to multiple sender nodes $A_1, A_2, \dots$. For each edge $A_i \rightarrow B$, we use binary values ‘0’ and ‘1’ to represent the activation state of the corresponding node. **For simplicity, we adopt the notation of the node itself to denote its activation state. [footnote4]** Specifically, $A_i = 0$ indicates that node $A_i$ is removed, ablated, or deactivated, ...”
> > >
> > > "**[footnote4]**{Reminded by the Reviewers ftP7, it is important to note that the concept of a “node” should be distinguished from that of a node’s activation state. Equations involving “node $B$” typically refer to high-dimensional representations such as activation values, whereas equations involving “the activation state of node $B$” concern the binary value (0/1) indicating whether the node is ablated, removed, or deactivated. Since this work does not address the specific form of activation values, we use the term “node $B$” as a shorthand for “the activation state of node $B$”.}"

---

> ### Author Response · Authors · 2025-08-03
> **Response**
>
> ### **Q2:**
> Thank you for your question. In fact, when we refer to “*any circuit*,” we specifically mean *any circuit obtained from Ns, Dn, or both*. Both Ns and Dn are constructed via ablation or knockout procedures to examine how the effect of a node  (or edge) on the output changes, and thereby induce the corresponding circuit graph. As a result, the nodes and edges derived from the circuit graph share a unified interpretation:
>
> - **Node**: A fundamental component that exerts an effect on the output. Depending on the level of granularity, nodes may correspond to attention heads, MLP modules, or finer components such as q, k, and v. This also includes coarser units such as the entire attention head, as you mentioned.
> - **Edge**: Indicates that a receiver node receives activation from a sender node. Likewise, if node $i$ does not emit an edge to node $j$, this implies that the output (activation or representation) of node $i$ is not part of the input to node $j$.
>
> Therefore, we would like to clarify a potential misunderstanding: an edge merely denotes the presence or absence of an activation pathway—it does **not** encode "*scalar-valued functions*" or "*functions with higher-dimensional output spaces*". For example, the edges $a5.8 \rightarrow m5$ and $a5.9 \rightarrow m5$ indicate that the inputs to the MLP in layer 5 include the outputs of the 8th and 9th attention heads, Moreover, other attention-to-MLP connections—such as $a5.0 \rightarrow m5$, $a5.1 \rightarrow m5$, etc.—are ablated. In this case, the MLP output is:
>
> $\text{output} = \text{MLP}(\text{residual stream}, a5.1.\text{corrupted output}, a5.2.\text{corrupted output}, ...,  a5.8.\text{clean output}, a5.9.\text{clean output}, ...).$
>
>
> Additionally, we recommend referring to **Section 2.4 of Paper[1]**, which discusses general patterns associated with AND and OR gates. Notably, the notion of "component" in that work is **agnostic to granularity** and is implicitly assumed to apply across all levels. In **Paper [2], Section 4.1** investigates OR gates where the fundamental unit is a *subcircuit*—a collection of neurons, typically **comprising multiple attention heads**. In contrast, **Appendix M of Paper [3]** examines OR gates at a finer granularity, such as **individual attention heads, embeddings, or MLPs**. These references collectively support our claim that "*circuits derived from either Ns or Dn, regardless of granularity, exhibit logical gate-like structures including AND, OR, and ADDER gates*".
>
> Accordingly, we have decided to revise the statement **“any circuit can be represented as a combination of AND, OR, and ADDER gates”** to the more precise formulation:
> **“In circuit graphs derived from Ns and Dn, the dependencies among nodes are mainly characterized by AND, OR, and ADDER gates.”**
>
> As we previously noted in our response to Q4, it is possible that other types of gates—such as XOR or NOT—may also be present. However, such gates cannot be identified via simple set intersections. We will explicitly acknowledge this limitation in the revised manuscript.
>
> **reference**
>
> *[1]. Heimersheim, Stefan, et al. "How to use and interpret activation patching." arXiv 2024.*
>
> *[2]. Wang, Kevin Ro, et al. "Interpretability in the Wild: a Circuit for Indirect Object Identification in GPT-2 Small." ICLR 2023.*
>
> *[3]. Conmy, Arthur, et al. "Towards automated circuit discovery for mechanistic interpretability." NeurIPS 2023.*

---

> > ### Comment · Reviewer_ftP7 · 2025-08-03
> >
> > Thank you for clarifying Q2--this helps and I agree the revised statement is much improved. However, it's still not clear to me why one should even expect most nodes to be of this type--can the authors justify this statement? If there are node types other than these three, then how do we know these three are jointly the dominant types?
> >
> > In thinking about this, I think I've realized I don't quite understand the ADDER gate. Since I now understand that these logical names just describe a certain property of the circuit nodes which matches that of their logical namesake, the ADDER gate seems a much weaker one because all it takes to be an ADDER gate is basically for ablation of an incoming circuit node to cause something between full and no ablation of the receiver circuit node. But (a) that's basically all possibilities except for 0 and 1, and (b), you haven't actually shown the set of ADDER gates you find satisfies this property, only that it satisfies this property *on average*. But even if you took a bunch of OR and AND nodes and averaged them, you'd get ADDER behavior, so it seems like the ADDER table you observe is almost tautological--if you average over nearly any set of nodes, unless they've been specifically selected to be OR or AND nodes (which the authors have already picked out), that they will exhibit behavior similar to the ADDER gate table, with ablation of one node, on average, having *some* effect. So it feels to me like the three node types the authors have identified may really be more like OR, AND, and THE_REST, and it's not clear how a more precise connection could even be established between the set of nodes claimed to be ADDER nodes and their logical counterparts, since their logical counterparts don't uniformly exhibit the same behavior when one sender node is set to 0, so I don't see how you could show that the circuit nodes labeled as such nearly exactly match that behavior (as you have shown for OR and AND nodes)

---

> ### Author Response · Authors · 2025-08-03
> **response**
>
> Thank you for your question. We would be glad to further clarify the rationale behind our definition of the **ADDER** gate.
>
> As discussed in your initial question (Q1), setting $B = 0$ does not necessarily imply that node $B$ has zero effect on the output. This effect is usually negligible but nonzero—for example, around 0.01. In all existing circuit discovery methodologies, some form of **thresholds** is employed to determine whether a node or edge has a meaningful contribution. For instance, in the ACDC algorithm applied to the IOI task, a threshold of 0.0575 is used: only those nodes and edges for which ablation causes $\Delta \mathrm{KL} > 0.0575$ are retained in the final circuit graph.
>
> We also adopt this threshold (0.0575) as the decision boundary for a node’s activation state in ACDC for IOI task. That is, if ablating an edge leads to an increase in output divergence such that $\Delta \mathrm{KL} > 0.0575$, we represent this influence by changing the state of the receiver node.
>
> The distinction between **AND** and **ADDER** gates lies in how they respond to multiple edge ablations. In the case of an AND gate, although ablating any single edge may lead to $\Delta \mathrm{KL} > 0.0575$, ablating *additional* edges no longer causes a further increase beyond this threshold. As a result, only two discrete states (0/1) are needed to represent the node’s behavior, which aligns with the logical semantics of an AND gate.
>
> In contrast, an ADDER gate exhibits **cumulative influence**: each edge ablation individually results in $\Delta \mathrm{KL} > 0.0575$, regardless of whether other edges have already been ablated. Thus, multiple discrete states (e.g., 0/1/2/...) are required to capture the progressive impact on the node’s output, justifying the use of the term **ADDER**.
>
> For all ADDER gates, the following condition holds: **for every edge, ablating it leads to $\Delta \mathrm{KL} > 0.0575$, independent of whether other edges have already been ablated**. This behavior is **algorithmically self-verified**.
>
> What is **algorithmically self-verified**?
>
> We can formalize this claim using a proof by contradiction. Suppose there exists an edge $A_2$ within an ADDER gate such that, when $A_1$ has already been ablated, ablating $A_2$ does **not** lead to $\Delta \mathrm{KL} > 0.0575$. In this case, under the **Ns algorithm**, which includes in the circuit graph only those nodes and edges with $\Delta \mathrm{KL} > 0.0575$, $A_2 \rightarrow B$ would not be included. However, the **Dn algorithm**, which replaces corrupted activations with clean ones, *would* include this edge. This implies that $A_2 \rightarrow B$ belongs to the set of edges that **appear in Dn but not in Ns** (i.e., OR gate set)—contradicting the assumption that $A_2$ is part of an ADDER gate, which by definition must be present in both algorithms.
>
> In summary, this self-verifying property of the algorithms ensures that:
> - **AND** and **OR** gates lead to exactly one instance of $\Delta \mathrm{KL} > 0.0575$, triggered when **all** edge states are 0 (AND gate) or when **any** edge state is 0 (OR gate).
> - **ADDER** gates produce **multiple** instances of $\Delta \mathrm{KL} > 0.0575$, one for each individual edge ablation.
>
> (Details regarding the algorithmically self-verified proof can be found in **Appendix A**, which provides a comprehensive account of the ablation (intervention) results for Ns​ and Dn​ across the three types of gates.)
>
> Finally, regarding your suggestion that the ADDER gate resembles what you call "The REST": we have considered this framing. However, we believe the term **ADDER** remains more appropriate, as each change in edge state corresponds to a proportional change in the node’s behavior—a pattern that strongly mirrors the additive logic of a real-world adder.

---

> ### Comment · Reviewer_ftP7 · 2025-08-04
>
> Thank you for the revision of Definition 1, I agree this helps clarify the overloaded notation.
>
> Regarding your reply about the ADDER gate, I realize that "0" doesn't quite mean zero, there needs to be some rounding, and of course I'm fine with that. In this comment when I say 0, I mean nearly zero (which I guess in your paper means \le 0.0575 in KL). I think the reviewers have confirmed that the ADDER gate is a catch-all for the gates which are not OR or AND gates that are found by Ns and Dn, and they prefer calling it an ADDER gate because of these gates' similar behavior to an ADDER gate. I guess I am fine with this if it's what the authors prefer, but it seems the behavioral similarity for these gates is substantially weaker than for the AND and OR gates, where the logical gates imply table values of all 0 or 100% and the categorized circuit gates match these. For the ADDER gates, all we know is that the table values for single ablation are non-zero, which still leaves a large range of values. In their comment's last sentence, the authors claim "each change in edge state corresponds to a proportional change in the node's behavior", which to me suggests that each edge in an ADDER gate has an (approximately) *equal* non-zero impact. But I don't think they have demonstrated this, and in the same comment they only claim that each edge just has a non-zero impact. If they could show these gates' incoming edges all had a nearly equal impact, I would find the ADDER gate analogy more appropriate (especially alongside the much stronger AND and OR analogies).
>
> Ultimately, the above is just presentational at this point I think, which is up to the authors as it is their paper. I appreciate their explanations and now feel I understand the paper's contribution much better. I think the authors present an interesting idea, but now that I understand these gates better I'm a bit less convinced of their interpretability impact, so I will raise my score to a 4 (I can't yet without submitting the "Final Justification" which requires having discussed with other reviewers and AC, which hasn't happened yet).

---

> > ### Author Response · Authors · 2025-08-05
> >
> > Thank you for your response and positive feedback. Regarding your final suggestion, we will carefully consider whether there might be a more effective way to explain ADDER. However, we would like to clarify one point: this work primarily focuses on interpreting faithfulness and completeness through the lens of logical gate patterns. Concerning these two metrics, ablating any edge in ADDER would, in theory, affect the optimal values of faithfulness and completeness. Therefore, from the perspective of this study, whether each edge contributes equally is less critical; what matters more is that each edge demonstrably impacts faithfulness and completeness.
> >
> > Once again, we sincerely appreciate your recognition of our work.

---

### Official Review · Reviewer_mrxZ · 2025-07-07

**Clarity:** 3
**Significance:** 3
**Originality:** 3
**Rating:** 5
**Confidence:** 3

**Summary:**

This paper presents an interesting abstraction of the behaviour that nodes and edges exhibit in circuits. In particular, it hypothesises that nodes are analogous to AND, OR, or ADDER logic gates in how they handle the input from incoming edges. The authors propose a framework that classifies nodes into one of the three gate classes based on their behaviour with respect to the intervention introduced its incoming edges. Using these identified behaviour, they were able to propose a more refined version of the completeness evaluation that could benefit the circuit discovery community.

**Questions:**

- Please refer to the Weaknesses section. I’d be interested to hear your thoughts on the issue of completeness.
- Are there any previous convention to abbreviate noising-based intervention as Ns and denoising-based intervention as Dn?

**Ethical Concerns:**

["NO or VERY MINOR ethics concerns only"]

**Final Justification:**

Thanks for the response. I am satisfied with the elaboration. Please incorporate them in the updated draft. I will maintain my positive assessment.

**Limitations:**

yes

**Quality:**

3

**Strengths And Weaknesses:**

### **Strength**

- This paper is very clearly written and easy to follow.
- It proposes a refinement to the completeness aspect of the evaluation of circuit discovery methods, an essential component that is often poorly assessed.
- The paper thoroughly enumerates its limitations, which is important for any interpretability research right now, as we are still in a very early stage of cracking the black box open. The paper does not overclaim anything and carefully states its findings. This should be much appreciated.

### **Weakness**
- I would like the paper to elaborate further on the limitations of existing completeness measures. Section 2.2 briefly mentions some critiques of sampling-based completeness scores, but the discussion is limited. It is more or less accepted that previous approaches to completeness evaluation are problematic, but, to the best of my knowledge, there has been no empirical or critical examination of this. I believe this work could have taken that step. It would also motivate their method strongly.

### **Smaller points:**
- The choice to abbreviate noising-based intervention as Ns and denoising-based intervention as Dn is somewhat unconventional.

---

> ### Author Rebuttal · Authors · 2025-07-28
>
> Thank you for your suggestion and recognition! We also agree with your perspective that the discussion of completeness is both cutting-edge and interesting. Below, we will engage in a discussion with you on the topic of completeness, in relation to existing work in this area.
>
> **### Q1: I would like the paper to elaborate further on the limitations of existing completeness measures. Section 2.2 briefly mentions some critiques of sampling-based completeness scores, but the discussion is limited. It is more or less accepted that previous approaches to completeness evaluation are problematic, but, to the best of my knowledge, there has been no empirical or critical examination of this. I believe this work could have taken that step. It would also motivate their method strongly.**
>
> Regarding the discussion of completeness, we conducted two experiments to address two key questions:
>
> **Key Question 1**: How can we demonstrate that the new evaluation metric $D(G \setminus C \,|G) $ performs better than the previous evaluation metric $D(C \setminus K \,\| G \setminus K)$?
>
> **Key Question 2**: How can we illustrate the potential contributions of completeness research to other areas of research?
>
> **Answer to Key Question1**
>
> We agree with your point *"there has been no empirical or critical examination of this"*. However, we must emphasize that the lack of empirical comparison arises from a fundamental **difference in the definitions** used by the two approaches. The earlier completeness evaluation method (proposed by Wang et al. [1]) in fact measures **incompleteness**, defined as$D(C \setminus K \,\| G \setminus K)$, where a lower score indicates better performance. In contrast, the current method [2] measures **completeness**, defined as $D(G \setminus C \,|G) $, where a higher score is preferable.
>
> Importantly, these two metrics are **not directly interchangeable** — that is,$1 - D(C \setminus K \,\| G \setminus K) \neq D(G \setminus C \,|G)$.
>
> Nevertheless, a comparison can still be made in terms of the **standard deviation**. It has been theoretically demonstrated that the earlier completeness method suffers from **high variance due to insufficient sampling**, leading to **instability** in the final results. To illustrate this, we provide a simple comparison on standard deviation between the previous and current evaluation methods, as shown in the table below:
>
> | Method | 5 sampled K|10 sampled K |20 sampled K |30 sampled K|New metric|
> | ------ | --------   |--------      |--------     |--------    |----|
> | ACDC  |  1.1±1.6    | 1.8±1.2      |1.4±1.1      |1.2±0.6     |1.3±0.13|
> | EAP   |  1.7±1.5    | 1.3±1.6      |1.6±1.2      |1.3±0.7     |1.4±0.11|
> | Edge-Pruning|  1.5±1.8    | 1.4±1.9      |1.4±1.6      |1.6±0.6     |1.2±0.08|
> | Ours  |  0.4±0.3    | 0.6±0.4      |0.3±0.3      |0.3±0.2     |1.8±0.03|
>
> We conducted five different runs of circuit discovery using random seeds, with the random subset $K$ chosen from subcircuits of sizes ranging from 2 to 5 nodes. For the previous evaluation metric, we performed 5, 10, 20, and 30 sampling iterations.
>
> The experimental results clearly support two conclusions:
>
> 1. For algorithms that inherently lack completeness (such as ACDC, EAP, and Edge-Pruning), as well as for our own method, which ensures completeness, the previous evaluation metric exhibits **large standard deviations** across all trials. This indicates that the sampling process for the previous metric is insufficient, rendering the results **highly unreliable**.
>
> 2. The new evaluation metric (see "New metric" column) **significantly reduces variance** for all types of circuits, as it does not suffer from sampling issues. The variance observed in the results arises from minor differences in the circuits generated by different random seeds.
>
> **reference**
>
> *[1]. Wang, Kevin Ro, et al. "Interpretability in the Wild: a Circuit for Indirect Object Identification in GPT-2 Small." ICLR 2023.*
>
> *[2]. Yu, Lei, et al. "Functional faithfulness in the wild: Circuit discovery with differentiable computation graph pruning." arXiv 2024*
>
> **Answer to Key Question2**
>
> We have designed a **toy task** analogous to **knowledge unlearning** to demonstrate the actual effectiveness of completeness. Specifically, given a harmful-response datasets (PKU-SafeRLHF), we first obtain its circuit $\mathcal{C}$, and then remove this circuit from the computation graph $\mathcal{G}$, observing whether the model can still respond to the dataset. The dataset provides harmful responses for certain questions, which are used to evaluate the efficacy of preventing the model from generating these harmful responses.
>
> In the table below, we present the performance of this toy task on the accuracy of $\mathcal{G-C}$, which refers to the performance of the computational graph after the circuit has been removed (using the interchange ablation method), reflecting the completeness of the circuit. We test two circuit discovery methods: EAP and Edge-Pruning.
>
> | Method | accuracy |accuracy of $\mathcal{G-C}$|
> | ------ | -------- |-------------------------- |
> | EAP   |  92.74   | 29.17                   |
> | Edge-Pruning|  92.74    | 31.44                   |
> | Ours  |  92.74    | 13.21                   |
>
> **Experimental Results**:
> It is evident that, compared to existing circuit discovery methods, our method ensures that the computational graph loses almost all the necessary mechanisms for the task once the circuit is removed. Additionally, this toy task highlights the significance of studying circuit completeness, which holds great potential in unlearning harmful information: **a complete circuit can help us pinpoint all neurons associated with harmful responses**.
>
> ### **Q2: The choice to abbreviate noising-based intervention as Ns and denoising-based intervention as Dn is somewhat unconventional.**
>
> Thank you for your suggestion; this is indeed somewhat unconventional. In previous works, *"noising-based intervention"* is often abbreviated as *"noising,"* and *"denoising-based intervention"* as *"denoising."* We abbreviated them as *"Ns"* and *"Dn"* to reduce the page load. In the revised version, we will replace these abbreviations with *"noising"* and *"denoising."*

---

> > ### Comment · Reviewer_mrxZ · 2025-08-03
> >
> > Thanks for the response. I am satisfied with the elaboration. Please incorporate them in the updated draft. I will maintain my positive assessment.

---

### Official Review · Reviewer_xbit · 2025-07-10

**Clarity:** 3
**Significance:** 2
**Originality:** 3
**Rating:** 4
**Confidence:** 3

**Summary:**

1. This paper proposes a new framework for interpreting circuits in language models by decomposing them into three types of logic gates: AND, OR, and ADDER. The author argues that most existing circuit discovery methods fail to identify complete circuits due to the incomplete recovery of OR gates.
2. This paper provides a detailed analysis showing the limitations of current noising-based and denoising-based strategies, and proposes a combined approach to recover logically complete circuits.
3. This paper provides extensive experimental validation using three different circuit discovery methods (ACDC, EAP, EdgePruning) on three tasks (IOI, GT, SA). The results show that the proposed Ns+Dn framework discovers circuits that are more complete and less random than circuits found using Ns alone, while maintaining comparable faithfulness.
4. This paper explores the functional roles these different gates play, suggesting AND gates integrate diverse functions while OR gates provide redundant pathways.

**Questions:**

Please see "Weaknesses".

**Ethical Concerns:**

["NO or VERY MINOR ethics concerns only"]

**Final Justification:**

I raised my score from 3 to 4 after the rebuttal. The authors addressed key concerns, including the AND/OR gate balance, practical value of completeness via a toy task, and clarified gate effects. Their plan to improve figures and add quantitative results is appreciated.

The main limitation (lack of evaluation beyond GPT2-small) remains, but is reasonable given current field constraints. Overall, the response strengthened the paper’s clarity and contribution, and my final rating is "4: Borderline accept".

**Limitations:**

Yes

**Quality:**

3

**Strengths And Weaknesses:**

Strengths:
1. The idea of decomposing a circuit into AND, OR, and ADDER gates to understand circuit functionality is a strong and meaningful contribution. It provides an intuitive and straightforward way to understand the relationships between components in a neural network.
2. The paper analyzes how the two primary intervention strategies (noising-based and denoising-based) are naturally suited to recovering different types of logic gates. The proposed framework (combine *Ns* and *Dn*) is modular and compatible with existing methods, making it practical for adoption.
3. Extensive experiments are conducted to support the paper's claims. The framework is tested across three different classes of circuit discovery algorithms on three distinct benchmark tasks.
4. The paper is clear and easy to follow.

Weaknesses:
1. One of the key parts of the method for separating the logic gates is based on the assumption that "*optimal alignment occurs when Ns and Dn contain an equal number of edges*" (L-231). This is a strong assumption that is not deeply justified. Is it possible that the true underlying circuit has an inherent imbalance between necessary (AND-like) and redundant (OR-like) pathways? How will the author handle this case? More discussions on this are suggested.
2. The "logically complete" circuits discovered by the proposed Ns+Dn method are not significantly more faithful than those found by standard Ns methods alone (as acknowledged by the authors in Sec 6.1). Given that faithfulness is often a key goal, this may limit the method’s practical appeal for users who prioritize that metric.
3. Some of the figures (e.g., Fig. 4 and 9) are difficult to understand.
(1) For example, the multi-line graphs in these 2 figures show two metrics across multiple methods on dual y-axes. It is visually dense and makes it challenging for reviewers to compare results effectively.
(2) Also, it seems to me (from Fig. 5) that KL divergence and task accuracy of Ns+Dn are very close to Ns-only. This raises my concern: why should we adopt this more complex method if existing approaches already perform nearly identically on core metrics? Is it possible to show the improvement through specific quantitative numbers rather than a line chart?
4. The paper refers to the "*gate effect*" as the contribution of an entire gate, which is then used to calculate the "*edge effect*" (e.g., Sec. H). However, the method for isolating and calculating the total effect of a multi-edge gate (e.g., for AND and OR gates where inputs are not independent) is not fully detailed. It is unclear to me how these contributions were measured, and more explanations on this are suggested.
5. All experiments are done on GPT2-small. There is no discussion showing whether the proposed Ns+Dn approach scales to other models (e.g., GPT-3, Llama).

---

> ### Author Rebuttal · Authors · 2025-07-29
>
> Thank you for your review! Your questions and suggestions have helped us improve the level of detail in the paper. Below, we provide point-by-point responses to each of your comments.
>
> ### **Q1: One of the key parts of the method for separating the logic gates is based on the assumption that "optimal alignment occurs when Ns and Dn contain an equal number of edges" (L-231). This is a strong assumption that is not deeply justified. Is it possible that the true underlying circuit has an inherent imbalance between necessary (AND-like) and redundant (OR-like) pathways? How will the author handle this case? More discussions on this are suggested.**
>
> Thank you for your suggestion. We discuss this issue in detail in Appendix E. As shown in Figure 8, when the OR set significantly exceeds the AND set, a large number of misclassified ADDER gates appear only within the OR set, leading to a much higher misalignment score for OR compared to AND. Similarly, when the size of the AND set significantly exceeds that of the OR set, a large number of ADDERs may be incorrectly classified as AND, thereby increasing the misalignment score of the AND set. As you mentioned, the misalignment score is more of a **"property,"** reflecting the optimal scale at which OR and AND sets should intersect.
>
> We determine the **optimal ratio** of AND to OR gates by **minimizing** the $(\text{misalignment scores of AND} +\text{misalignment scores of OR})$. The optimal results obtained with different methods and datasets are approximately as follows:
>
> **Table: The optimal ratio of AND and OR sets**
>
> | strategies| IOI     | GT      | SA    |
> |--------   |-------- |-------- |-------|
> | ACDC      |1.12: 1  |1.05: 1  |1.07: 1|
> | EAP       |1.09: 1  |1.03:1   |1.05: 1|
> | Edge-Pruning|1.11: 1|1.07: 1  |1.06: 1|
>
> The results show that the proportion of AND is always **slightly higher** than that of OR, but it can be **approximated as 1:1**. This suggests that, in general, there is **no significant imbalance between the AND and OR sets**. This forms the basis for our conclusion that "*optimal alignment occurs when Ns and Dn contain an equal number of edges.*" Moreover, we will **add these discussion into Appendix E in the future version**.
>
> ### **Q2: The "logically complete" circuits discovered by the proposed Ns+Dn method are not significantly more faithful than those found by standard Ns methods alone (as acknowledged by the authors in Sec 6.1). Given that faithfulness is often a key goal, this may limit the method’s practical appeal for users who prioritize that metric.**
>
> Thank you for your comment. We fully agree with your perspective that faithfulness warrants significantly more attention than completeness. Nevertheless, we aim to explore potential applications from the perspective of completeness, in order to highlight its practical significance.
>
> We have designed a **toy task** analogous to **knowledge unlearning** to demonstrate the **actual effectiveness of completeness**. Specifically, given a harmful-response datasets (PKU-SafeRLHF), we first obtain its circuit $\mathcal{C}$, and then remove this circuit from the computation graph $\mathcal{G}$, observing whether the model can still respond to the dataset. The dataset provides harmful responses for certain questions, which are used to evaluate the efficacy of preventing the model from generating these harmful responses.
>
> In the table below, we present the performance of this toy task on the accuracy of $\mathcal{G-C}$, which refers to the performance of the computational graph after the circuit has been removed (using the interchange ablation method), reflecting the completeness of the circuit. We test two circuit discovery methods: EAP and Edge-Pruning.
>
> | Method | accuracy |accuracy of $\mathcal{G-C}$|
> | ------ | -------- |-------------------------- |
> | EAP   |  92.74   | 29.17                   |
> | Edge-Pruning|  92.74    | 31.44                   |
> | Ours  |  92.74    | **13.21**                   |
>
> **Experimental Results:**
>
> It is evident that, compared to existing circuit discovery methods, our method ensures that the computational graph loses almost all the necessary mechanisms for the task once the circuit is removed. Additionally, this toy task highlights the significance of studying circuit completeness, which holds great potential in unlearning harmful information: **a complete circuit can help us pinpoint all neurons associated with harmful responses**.
>
> ### **Q3: Some of the figures (e.g., Fig. 4 and 9) are difficult to understand. (1) For example, the multi-line graphs in these 2 figures show two metrics across multiple methods on dual y-axes. It is visually dense and makes it challenging for reviewers to compare results effectively. (2) Also, it seems to me (from Fig. 5) that KL divergence and task accuracy of Ns+Dn are very close to Ns-only. This raises my concern: why should we adopt this more complex method if existing approaches already perform nearly identically on core metrics? Is it possible to show the improvement through specific quantitative numbers rather than a line chart?**
>
> Thank you for your suggestion. To avoid making the figures too complex, we will **include a table with specific quantitative numbers in revised versions**. Due to the character limit, we are unable to provide detailed quantitative results in this response. However, we would be happy to share a table of the specific results during the subsequent discussion phase, should you be interested. Additionally, we provide a detailed explanation in **Q2** regarding the **significance of studying completeness**. In short, completeness offers potentially valuable insights for downstream tasks such as model unlearning.
>
> ### **Q4: The paper refers to the "gate effect" as the contribution of an entire gate, which is then used to calculate the "edge effect" (e.g., Sec. H). However, the method for isolating and calculating the total effect of a multi-edge gate (e.g., for AND and OR gates where inputs are not independent) is not fully detailed. It is unclear to me how these contributions were measured, and more explanations on this are suggested.**
>
> Thank you for your question. We will provide a specific explanation of the different effects:
>
> - **Edge effect**: The $\Delta KL$ resulting from ablating this edge.
> - **Gate effect**: The $\Delta KL$ resulting from ablating all the edges contained within this gate.
> - **Total effect of a multi-edge gate**: This is the same as the gate effect.
>
> Therefore, regardless of whether the input are independent, the **gate effect** of a given gate can be obtained **by ablating its receiver node**. In **Appendix H**, we further ensure that all gates used for effect computation have receiver nodes that are **not part of any branch of an OR gate**. This is because if a gate resides on a branch of a higher-level OR gate, then any ablation of that gate tends to have negligible impact on the final output. Additionally, this phenomenon is theoretically discussed in **Footnote 2 on Page 4**.
>
> Finally, in the **revised version**, we will **include these experimental details** on the local gate effect in Appendix H.
>
> ### **Q5: All experiments are done on GPT2-small. There is no discussion showing whether the proposed Ns+Dn approach scales to other models (e.g., GPT-3, Llama).**
>
> Thank you for your suggestion. However, current mainstream circuit discovery algorithms have been comprehensively studied only on GPT2-small. For instance, methods such as **ACDC**, **EAP**, and **Edge-Pruning** have **all** been developed and evaluated specifically on **GPT2-small**. Scaling these investigations to models with larger parameter sizes is itself a central and ongoing challenge in the field of circuits. We would be grateful for your understanding of this limitation.

---

### Official Review · Reviewer_khCH · 2025-07-10

**Clarity:** 2
**Significance:** 3
**Originality:** 3
**Rating:** 4
**Confidence:** 3

**Summary:**

This paper proposes a circuit discovery method using AND, OR, and ADDER gates. Categorizing existing intervention methods for circuit discovery as noising-based (Ns) and denoising-based (Dn), the authors argue that Ns recovers AND and ADDER gates while Dn recovers OR and ADDER gates. Furthermore, for optimal faithfulness, the circuit needs only one edge from each OR gate, while for optimal completeness it needs only one edge from each AND gate. The authors then propose to combine noising- and denoising-based interventions to discover all three types of logic gates. The proposed framework is evaluated through experiments on GPT2-small considering three tasks (indirect object inference, greater-than, and syntactic agreement). In particular, they show that the proposed Ns+Dn achieves higher completeness (measured by KL divergence and accuracy) than Ns only.

**Questions:**

1. Can there exist gates other than AND, OR, and ADDER, and would that affect Corollaries 1 and 2? I would also suggest adding a short proof sketch for them in the main text even though the full proofs are in the appendix.

2. Regarding how optimizing for faithfulness allows removing an edge from OR gates, while only one edge being present will activate the OR gate, it doesn’t necessarily mean that each input activates all edges into the OR gate. In the extreme case, if the inputs that activate $A_1$ are disjoint from those that activate $A_2$, then removing either from gate $B = A_1 \vee A_2$ will hurt faithfulness. Could the authors clarify why removing an edge in this case is still ok?

3. When running Ns and Dn in parallel to identify the three types of gates, are there any assumptions such as them having to use the same strategy (greedy, linear estimation, ..)? Does misalignment vary depending on which method was used?

4. In Section 4.3.2, the comparison should also include Dn methods especially because they aim to optimize for completeness.

Other questions/comments:
- Is the term “logically complete circuits” used with a precise meaning? I think this might not be the best name, potentially confused with completeness.
- I don’t quite follow how the paragraph starting in line 171 is implied by Corollary 2.
- Line 241: “Expanding the space of intervention combinations renders circuit discovery an NP problem”. Is this referring to it being an NP-hard problem? There should be a proof for the statement.
- The marks and legends in Figure 4 were too small and very hard to read.
- Figure 5: is there a statistically significant difference between Ns+Dn and Ns? I would hope to see error bars.

**Ethical Concerns:**

["NO or VERY MINOR ethics concerns only"]

**Final Justification:**

The authors have addressed many clarity issues and provided additional experiments to strengthen the evaluation. While I think the planned revisions will significantly improve the paper, the rebuttal also clarified that the definitions of gate types are quite specific to the noising- and denoising-based interventions introduced by this paper. This somewhat limits the completeness and significance of the proposed circuits, hence my keeping the score at 4.

**Limitations:**

Yes

**Paper Formatting Concerns:**

No major formatting issues.

**Quality:**

3

**Strengths And Weaknesses:**

The proposed approach to circuit discovery using logical gates is novel and interesting. Circuits with multiple, clearly-defined types of gates can lead to richer interpretations of language models.

The proposed method for discovering these nodes, by combining noising- and denoising-based strategies, is fairly straightforward and can be adopted by tweaking existing Ns methods. Thus, there are immediately a few variants of Ns+Ds for logical circuit discovery based on which baseline was chosen. This adds to usability and potential impact.

My main concern of the paper is that some definitions or results are described not quite rigorously, but rather using high-level arguments using intuition. This made it difficult to check technical correctness and significance: e.g., proofs for Corollary 1 and 2 and how the Ns+Dn objective in Eq 3 can be optimized using different baseline strategies. Also, faithfulness and completeness are defined in Section 2.2 such that there are multiple ways to quantify them, and I wasn’t sure if the proof for Corollary 2 still applies if faithfulness is measured by accuracy for example.

The authors compare their method mainly to Ns methods, with the reasoning that most existing baselines are noising-based. However, Table 1 shows Dn methods as well (although I am confused that the methods for Ns and Dn are identical), and it would be appropriate to compare against them wrt completeness.

---

> ### Author Rebuttal · Authors · 2025-07-29
>
> We sincerely appreciate your review and recognition of our work. Below, we provide detailed responses to each of the issues you have raised. Due to character limitations, some general concerns that were also noted by other reviewers may not be fully explained here. For a more comprehensive explanation of those points, we kindly refer you to the discussion pages associated with the other reviewers. We hope for your understanding in this regard.
>
> ### **Q1: Can there exist gates other than AND, OR, and ADDER, and would that affect Corollaries 1 and 2? I would also suggest adding a short proof sketch for them in the main text even though the full proofs are in the appendix.**
>
> Thank you for your question. As discussed in Section A2, all the gates under consideration are derived from the intersection between Ns and Dn. The three gates are **not high-level arguments derived from intuition**; rather, they **constitute a complete set that spans the intersection of Ns and Dn**. The union of these three gates encompasses all the nodes and edges present in both Ns and Dn , and therefore, we may reasonably assume that they are complete.
>
> Moreover, we must acknowledge that there **may be gates, such as the XOR gate, that the Ns and Dn frameworks cannot discover**. We will **include a discussion** on potentially undiscovered gates in the **limitations section** in the **revised version**; however, these gates would certainly require a circuit framework beyond Ns and Dn, which falls **outside the scope of the current research**.
>
> We will also incorporate **these explanations** and **a short proof sketch** into the main text in the **revised version** to support the discovery of the three gates.
>
>
>
> ### **Q2: Regarding how optimizing for faithfulness allows removing an edge from OR gates, while only one edge being present will activate the OR gate, it doesn’t necessarily mean that each input activates all edges into the OR gate. In the extreme case, if the inputs that activate A1 are disjoint from those that activate A2, then removing either from gate B=A1 or A2 will hurt faithfulness. Could the authors clarify why removing an edge in this case is still ok?**
>
> Thank you for your question. In the extreme case you mentioned, when $A1$ is removed, $A2$ will **still provide the same effect** for $B$, even though the input to $A2$ differs from that of $A1$. Additionally, both **Appendix M of paper [1]** and **Section 4.1 of paper [2]** indicate that removing an edge from an OR gate does **not affect faithfulness**. These two papers are foundational works in circuit discovery, and we believe they support the validity of our inference. Furthermore, you can refer to **Appendices A and B** for a detailed derivation of the OR gate.
>
> To further validate these conclusions, we removed one edge from each of the three gates obtained using the EAP algorithm on IOI, and observed the changes in faithfulness and completeness. The metrics for faithfulness and completeness were defined as in Section 2.2. The table below shows the average results on 30 randomly selected nodes:
>
> | gate                | faithfulness  |completeness|
> | ----------------    | ------------ |----------   |
> | AND                 |      0.43    |   0.38      |
> | AND removing 1 edge |      0.03     |   0.36     |
> | OR                  |      0.29    |   0.44      |
> | OR removing 1 edge |      0.28     |   0.19     |
> | ADDER                |      0.55    |   0.49      |
> | ADDER removing 1 edge |      0.23     |   0.24     |
>
> This table also demonstrates that removing one edge from the AND gate has a negligible impact on completeness, but it significantly reduces faithfulness. Similarly, removing one edge from the OR gate has almost no effect on faithfulness, but it severely reduces completeness. **More detailed results** about removing edges can be shown in **Q1 for Reviewer ftP7** (the last reviewer in this paper).
>
> **reference:**
>
> *[1]. Towards Automated Circuit Discovery for Mechanistic Interpretability, NeurIPS 2023*
>
> *[2]. Interpretability in the wild: a circuit for indirect object identification in gpt-2 small, ICLR 2023*
>
>
> ### **Q3: When running Ns and Dn in parallel to identify the three types of gates, are there any assumptions such as them having to use the same strategy (greedy, linear estimation, ..)? Does misalignment vary depending on which method was used?**
>
> Thank you for your question. We recommend using the **same strategy** for both Ns and Dn, as different strategies may not lead to the discovery of the corresponding ADDER gate. For instance, greedy search selects based on the order of searching, whereas EAP selects based on score ranking. Consequently, different strategies can also result in variations in misalignment.
>
> ### **Q4:  In Section 4.3.1, the comparison should also include Dn methods especially because they aim to optimize for completeness**
>
> Thank you for your suggestion! We will **include the experimental results for Dn in revised versions**. Due to character limitations, we could present the specific performance for Dn in the subsequent discussion.
>
> In brief, our method generally **outperforms** the approach based on Dn. Although the Dn-based method is theoretically capable of identifying all OR gates, it **suffers from certain shortcomings in practice**. For instance, as shown in Table 1, EAP fails to identify one of the edges associated with an AND gate under the Dn framework. Additionally, the greedy strategy may miss some OR gates located beneath branches of AND gates due to its dependence on the search order. The differentiable mask approach, on the other hand, suffers from the drawback that, in gates involving multiple edges, the mask values assigned to each edge tend to be lower, reducing its effectiveness.
>
> ### **Q5:  Is the term “logically complete circuits” used with a precise meaning? I think this might not be the best name, potentially confused with completeness.**
>
> Thank you for your suggestion. We are considering using the term "logically sound circuit" instead of "logically complete circuit." We would appreciate if the reviewer has a better suggestion.
>
> ### **Q6: I don’t quite follow how the paragraph starting in line 171 is implied by Corollary 2.**
>
> Thank you for your question. In the proof presented in **Appendix B** and the **table in A2**, we derived the following two conclusions:
>
> - As long as one edge of the OR gate is activated, the faithfulness of the circuit will not be changed.
> - As long as one edge of the AND gate is activated, the completeness of the circuit will not be changed.
>
> Therefore, we further conclude in the paper that "*faithfulness refers to the sum of all gate effects in the circuit, which should be as large as possible.*" In other words, **faithfulness** requires that **all gates in the circuit be activated**: for AND and ADDER gates, this means all incoming edges must be active; for OR gates, at least one edge must be active. Similarly, **completeness** requires that none of the gates remain active in $G \setminus C$. Therefore, in the circuit $C$, all edges of OR and ADDER gates must be activated, while for AND gates, activation of at least one edge is sufficient.
>
>
> ### **Q7: Line 241: “Expanding the space of intervention combinations renders circuit discovery an NP problem”. Is this referring to it being an NP-hard problem? There should be a proof for the statement.**
>
> Thank you for your suggestion. Due to the differences in circuit algorithms, this issue cannot be fully proven mathematically. However, we will provide some additional explanation: we treat **each ablation** as $O(1)$. Assuming the language model contains $N$ nodes, if we need to use all possible intervention combinations to obtain a complete circuit, the time complexity would be $O(2^N)$, which far exceeds polynomial time. However, **different strategies can optimize the solution verification process to varying degrees**. For example, EAP only requires $O(N)$ time to **verify** whether the circuit is complete, while Edge-pruning involves **gradient descent**, making its time complexity **difficult to estimate**. Additionally, greedy search can only perform a **greedy verification** in $O(N)$ time rather than a thorough verification. Therefore, we ultimately classify this problem as **NP problem**.
>
> **In the revised version, we will incorporate these analyses into the main text to provide a simple explanation for the "NP problem."**
>
> ### **Q8: The marks and legends in Figure 4 were too small and very hard to read.**
>
> Thank you for your suggestion and we will **replace these figures with tables in revised version**. Due to character limitations, we could present the specific performance for Dn in the subsequent discussion.
>
> ### **Q9: Figure 5: is there a statistically significant difference between Ns+Dn and Ns? I would hope to see error bars.**
>
> Thank you for your suggestion. We will add their standard deviations and now we show the std. of KL below:
> | strategies| 100     | 200     | 500     | 1000    | 2000 | 5000 |
> |--------   |-------- |-------- |-------- |-------- |----  |----  |
> | Ns        |2.72±0.12|2.51±0.09|0.61±0.06|0.51±0.03|0.44±0|0.41±0|
> | Dn        |2.96±0.16|2.91±0.15|0.95±0.07|0.62±0.06|0.53±0.01|0.49±0|
> | Ns+Dn     |2.57±0.11|2.26±0.10|0.55±0.04|0.42±0.01|0.42±0|0.40±0|

---

> > ### Comment · Reviewer_khCH · 2025-08-07
> >
> > Thanks for your response. These answers and your discussion with Reviewer ftP7 clarified the definitions of gates for me--I indeed had a similar misunderstanding. It appears that the gate types characterize how a gate behaves under ablations/interventions, rather than describing strict logical relationships. As a result, their definitions are quite closely tied to noising- and denoising-based techniques. I still think the ideas in the paper are good, and characterizing and discovering different gate types are interesting for circuit discovery; however, I think the presentation needs to be improved significantly to not misrepresent the ideas for readers.

---

> > > ### Author Response · Authors · 2025-08-08
> > >
> > > We sincerely appreciate your feedback and suggestions. Refining our definitions to avoid misleading the reader has been one of our primary concerns. To date, we have implemented the following revisions to the manuscript:
> > >
> > > 1. **Definition 1** — We have clarified the distinction between the *activation state* and the *activation value* of a node. Specifically, circuit gates indicate which combinations of edges influence the final output effect, rather than directly reflecting the intrinsic functions of the components themselves. (The updated definition can be found in the discussion with *ftP7*.)
> > >
> > > 2. **Limitations** — We have introduced a hypothesis concerning the potential existence of additional gates, while explicitly noting that, due to the limited detection capabilities of $N_s$ and $D_n$, such gates fall beyond the scope of the present study. We plan to explore alternative approaches for their discovery in future work. (A detailed explanation can be found in the discussion with *ftP7*.)
> > >
> > > 3. **Impact** — We have incorporated a discussion of the advantages of *completeness* in downstream unlearning tasks, thereby demonstrating the broader value of investigating completeness. (The corresponding results are presented in the discussion with *mrxZ*.)
> > >
> > > 4. **Proofs** — We will add a explanation for the conclusion that the problem is NP, among other results. (This addition corresponds to *Q7* in your discussion.)
> > >
> > > 5. **Additional Experiments** — These include:
> > >    - the precise ratio of the *misalignment score* (as discussed in *Q1* with reviewer *xbit*),
> > >    - the performance of $D_n$ in completeness-related experiments (as discussed in *Q4* with you, and the **detailed results are seen at the bottom of this response**), and
> > >    - enlarged and clearer font sizes for *Figure 4* and *Figure 5*.
> > >
> > > We believe these revisions address the reviewers’ concerns regarding potential ambiguities or difficulties in understanding our work. Should you have any further suggestions, we would be pleased to engage in additional discussion and make further refinements.
> > >
> > > Finally, we once again express our gratitude for your recognition of our work.
> > >
> > > **Appendix:**
> > >
> > > Completeness in IOI task with Ns, Dn, and Ns+Dn
> > >
> > > **KL**
> > >
> > > | #edges | ACDC_NS   | ACDC_Dn   | ACDC_NsDn |  EAP_Ns   |  EAP_Dn   |  EAP_NsDn |  EdgeP_Ns |  EdgeP_Dn |  EdgeP_NsDn|
> > > | ------ | ----------| ----------| ----------| ----------| ----------| ----------| ----------| ----------| ----------|
> > > |   100  |0.62±0.1   |0.64±0.1   |0.64±0.1   |0.51±0.1   |0.71±0.1   |0.74±0.1   |0.56±0.1   |0.62±0.1   |0.64±0.1    |
> > > |   200  |0.64±0.1   |0.69±0.1   |0.71±0.1   |0.56±0.1   |0.73±0.0   |0.73±0.1   |0.58±0.1   |0.65±0.1   |0.69±0.1   |
> > > |   500  |0.71±0.1   |0.84±0.1   |0.87±0.1   |0.65±0.1   |0.81±0.1   |0.83±0.1   |0.67±0.1   |0.88±0.1   |0.94±0.1   |
> > > |   1000 |0.81±0.1   |0.96±0.1   |1.03±0.1   |0.74±0.1   |0.96±0.1   |1.02±0.1   |1.08±0.1   |1.11±0.1   |1.13±0.1  |
> > > |   2000 |1.05±0.1   |1.19±0.2   |1.22±0.1   |0.89±0.1   |1.11±0.2   |1.24±0.2   |1.31±0.2   |1.40±0.1   |1.41±0.1  |
> > > |   5000 |1.34±0.2   |1.42±0.2   |1.46±0.2   |1.21±0.1   |1.42±0.2   |1.48±0.2   |1.62±0.2   |1.73±0.1   |1.73±0.1 |
> > >
> > > **accuracy**
> > >
> > > | #edges | ACDC_NS   | ACDC_Dn   | ACDC_NsDn |  EAP_Ns   |  EAP_Dn   |  EAP_NsDn |  EdgeP_Ns |  EdgeP_Dn |  EdgeP_NsDn|
> > > | ------ | ----------| ----------| ----------| ----------| ----------| ----------| ----------| ----------| ----------|
> > > |   100  |0.73±0.1   |0.68±0.0   |0.63±0.1   |0.74±0.1   |0.53±0.0   |0.51±0.0   |0.76±0.1   |0.67±0.1   |0.66±0.0    |
> > > |   200  |0.65±0.0   |0.62±0.0   |0.60±0.0   |0.68±0.0   |0.49±0.0   |0.46±0.0   |0.59±0.0   |0.55±0.0   |0.56±0.0   |
> > > |   500  |0.53±0.0   |0.51±0.0   |0.51±0.0   |0.61±0.0   |0.41±0.0   |0.38±0.0   |0.55±0.0   |0.43±0.0   |0.43±0.0   |
> > > |   1000 |0.40±0.0   |0.34±0.0   |0.33±0.0   |0.39±0.0   |0.29±0.0   |0.26±0.0   |0.32±0.0   |0.27±0.0   |0.26±0.0  |
> > > |   2000 |0.34±0.0   |0.26±0.0   |0.24±0.0   |0.35±0.0   |0.18±0.0   |0.14±0.0   |0.24±0.0   |0.18±0.0   |0.16±0.0  |
> > > |   5000 |0.26±0.0   |0.22±0.0   |0.21±0.0   |0.29±0.0   |0.13±0.0   |0.09±0.0   |0.21±0.0   |0.11±0.0   |0.11±0.0 |

---

### Official Review · Reviewer_mnpW · 2025-07-18

**Clarity:** 2
**Significance:** 3
**Originality:** 4
**Rating:** 2
**Confidence:** 4

**Summary:**

When finding which subgraph (circuit) of the NN is responsible for a task, we want to have three properties: faithfulness (the circuit by itself does the task), minimality (the circuit is as small as possible) and completeness (the circuit contains all parts of the NN that are responsible by the task, even redundant ones).

Circuit discovery algorithms until now focused mainly on faithfulness and minimality and the tradeoff between them. The circuits discovered by them often lack completeness. The "automatic circuit discovery" paper [ref 2 in paper, appendix M] also talked about a logical OR gate as something that its proposed method had trouble with, and methods since has not improved.

This paper argues that traditional circuit discovery algorithms miss OR gates because they use a noising (Ns) approach. Ns algorithms start with NN activations on 'clean' inputs, and replace them with 'corrupted' activations; which leads them to see that replacing one if the inputs is OK (so it has no effect).

The paper introduces denoising (Dn) as an alternative circuit discovery approach: modify any current algorithm to start with the corrupted activations, and replace parts of it with clean activations. The same way that Ns algorithms easily discover AND gates, and AND is the dual of OR (DeMorgan's law style, see lines 199-201), Dn algorithms easily discover OR gates.

To make an algorithm which discovers both kinds of gates (as well as linear ADDER gates), we can combine Dn and Ns approaches in the same loss.

**Questions:**

# Upgrading my score
When the authors address all the weaknesses I wrote about, I will recommend this paper for acceptance.

# Questions about experiments
This section contains some clarity problems that should be addressed

## Section 4.1
- In section 4.1, how do you choose the C_AND, C_OR and C_ADDER circuit parts? Using *which* Dn and Ns algorithm specifically? On how many random runs?
- How do you choose the 1 and 2 edges to remove for Figure 3? Randomly, you average over all sets of 1 edges and sets of 2 edges, ... ?
- I think the "granularity alignment" concept from Figure 2 and Lines 222-233 needs more explanation. Why is Figure 2 a good picture of what's really happening in the circuit? The misalignment scores from Appendix E, they're completely separate for OR and AND; from figure 2 I thought misalignment was a property of the ADDER intersection? Very confused here.

## Completeness validation 1: removing the circuit

This corresponds to Section 4.3.1 (in page 7) and Figure 4

Figure 4 is really hard to read, the dots are small and everything is the same color, so I don't know what to make of the results at all. In any case, the difference between circuit discovery methods doesn't seem that large? But it's consistent at least.

From the text in lines 265-279, it seems that you're measuring completeness by removing the discovered circuit from the model and checking how much the KL divergence changes. Is that true?

If so, that doesn't match the definition of completeness from Wang et al. 2023: that's about whether removing a *subcircuit* K from both the model and the circuit, has the same effect on both.(as you well specify in lines 118-124). Why is this method a valid way to measure completeness?

## completeness verification 2: average Hamming distance
What is the Hamming distance on: which edges are present/absent? Is this the average of every pair of circuits from the 30 runs (900 pairs total), or from a few randomly selected pairs?

It would be much easier to read Table 3 if you would just compute the Hamming distance for C_Ns and for C_{ns + dn} and state both separately,  instead of subtracting them. Then we can compare which distance is greater.

## Figure 6 + section 5.1

How are you supporting the claim that "the AND gates typically receive edges from different functions" vs "the OR gates receive edges from the same function"? What is a function here? Do you literally mean whether incoming edges are from MLPs ("same function") or attentions and MLPs ("different function")? That's a very strange thing to check, different MLPs and different attention heads will be computing different function, so I don't see why they're the same

# Limitations that aren't actually limitations
In Line 336-338, you say that "since there is no difference in faithfulness between the logically complete circuit and the existing circuit, it is difficult to directly demonstrate its 'superiority' over current work". This seems false: you can just demonstrate superiority by demonstrating that the discovered circuit is more complete. (which the Hamming distance experiment kind of showed). Why did you say this?



# Minor awkward wording
L46: "we introduce the concept of logic gates" - no, you do not, it's at least a century old. You talk about logic gates in circuits.

**Ethical Concerns:**

["NO or VERY MINOR ethics concerns only"]

**Final Justification:**

Given conceptual errors in the paper (presenting OR, AND and ADDER as fundamental whereas they come from the framework of Noising and Denoising that they just introduced; and lack of consideration in the argument to NONE gates that are detected by neither Noising nor Denoising) I'm keeping my score.

**Limitations:**

The one limitation they acknowledge, in lines 335-340, that this is only theoretical research and cannot be shown empirically to be better, is straight up not a limitation.

**Quality:**

1

**Strengths And Weaknesses:**

# Strengths

This paper engages reasonably well with the existing literature. Its main idea, to introduce Denoising as an approach, is excellent: it makes it obvious everyone was making an assumption that restricted the design space. The resulting design plausibly addresses the largest remaining within-paradigm weakness of circuit discovery (OR gates).

I really like the conceptual clarity of Table 1. In contrast with the attempted formal Definition 1 of the article, Table 1 very clearly defines what we expect of "redundant + voting" circuits (OR) or "all outputs must work" circuits (AND), or linear (ADDER) circuits.

The end result is a brilliant idea that, if I were writing a circuit discovery paper, I would definitely try out and give credit to.

# Weaknesses

Unfortunately, the paper suffers from two issues: superficial empirical validation, and conceptual problems.

## Empirical problems.

There is not enough grounds for readers to believe that the presented method is actually better than the existing ones, despite the clearly motivated idea. A

## Conceptual problems

### Correctness of OR/AND/ADDER definitions

Definition 1 (L133-143) introduces OR, AND and ADDER gates with seemingly formal definitions, but these are very unclear. what is the input space to which the As and the Bs belong to? If it is booleans, what does the sum (+) operation mean? If they are numbers, what do OR (v) and AND (^) mean?

What reason do we have to believe that NN circuits can be decomposed into these three gates? Neural networks do other operations as well, for example negation (if we stick to the boolean framing, which I also think is too restrictive).

It seems that the real definition of OR, AND and ADDER is whether they can be discovered by linear approximation methods or existing methods in Ns or Dn versions (L196-206 in Section 4). I think you should stick to that.

### It's not true that you can just remove edges from OR and AND gates, or that OR gates

In the caption of figure 1: "when optimizing for faithfulness and sparsity alone, it is possible to remove one edge from the OR gate". Why? In the subset of inputs in which the node corresponding to the removed edge is activated, and the edge that remains is not, this changes the output of the circuit. So removing one edge *does* decrease faithfulness, which would make existing algorithms find the OR gates sometimes.

### It's not true that all circuits can be decomposed in OR, AND and ADDER (as defined)

Line 8-9: "decompose the circuit into combinations of [OR, AND and ADDER]" -- which circuit do you decompose in this way? You mean *any* circuit? How do we know that any circuit can be expressed with these three gates? (This claim is made again in L46 and L153-155).

How to fix?

If you change the definition to "AND can be discovered by Ns, OR by Dn, and ADDER by both" then you would just have to show that the OTHER gates (which can be discovered by neither Dn or Ns) don't exist. Then, every circuit is composed of these gates (note, I'm not sure that's true).

You could also argue that all circuits are OR and AND and NOT gates if they're made of booleans, but that seems like a huge stretch given that neural networks inputs and outputs are continuous.

## Missing literature

In line 103, these two papers are missing for linear estimation: EAP-IG (https://arxiv.org/abs/2403.17806) and AtP* (https://arxiv.org/abs/2403.00745). Both of these are reasonably cited elsewhere, especially EAP-IG. They address some of the issues of EAP/AtP for circuit attribution.

---

> ### Author Rebuttal · Authors · 2025-07-29
>
> We sincerely thank you for your thorough review of our manuscript. However, due to the character limit imposed on this rebuttal, we are currently only able to provide responses with as much detail as possible, though some quantitative results may not be fully presented at this stage. In addition, some of our responses are necessarily too detailed and lengthy to be included in full here. We kindly ask you to refer to the responses provided to other reviewers, as guided in our navigation, for a more complete discussion. We hope for your understanding in this regard and look forward to the opportunity for subsequent discussions. Below, we address your comments one by one:
>
> ### **Q1: There is not enough grounds for readers to believe that the presented method is actually better than the existing ones, despite the clearly motivated idea.**
>
> Thank you for your question. While our method offers only a marginal improvement over existing approaches in terms of faithfulness, it significantly outperforms them in terms of completeness. Regarding the question of **"how to demonstrate the practical value of completeness,"** we have addressed this point **in our response to Reviewer mrxZ’s Q1** (The **fourth reviewer in this paper**, see **Key Question 2: How can we illustrate the potential contributions of completeness research to other areas of research?**).
>
> In brief, we validate the core concept of our work—**circuit completeness**—through a downstream task in **model unlearning**. Specifically, we show that **completeness can offer indirect but meaningful support for model unlearning**, particularly in the context of removing harmful responses. This is a capability not afforded by existing methods that focus primarily on faithfulness.
>
> ### **Q2: Correctness of OR/AND/ADDER definitions: Definition 1 (L133-143) introduces OR, AND and ADDER gates with seemingly formal definitions, but these are very unclear.**
>
> Thank you for your question and suggestions. We recognize the need to further clarify **how a circuit gate transitions from the continuous activation values of neurons to discrete states such as 0, 1, or 2**. This issue involves numerous experimental results and nuanced explanations, and we kindly refer you to our **response to Reviewer ftP7’s Q1** (The **fifth reviewer in this paper**) for a more detailed discussion.
>
> Briefly, we interpret the state of an edge or node as follows: a state of **1 denotes active, retained, or exerts an effect**, while a state of **0 indicates inactive, ablated, or has no effect**. When minor variations in effect are ignored, certain subcircuit patterns exhibit **truth tables identical to those of AND, OR, or ADDER gates**. Thus, we adopt the terminology of AND, OR, and ADDER gates to describe these subcircuits.
>
> ### **Q3:  It's not true that you can just remove edges from OR and AND gates, or that OR gates: In the caption of figure 1: "when optimizing for faithfulness and sparsity alone, it is possible to remove one edge from the OR gate". Why?**
>
> Thank you for your question. Through **A2**, I believe we can clarify this misunderstanding. Ablating an OR edge （edge1=1 and edge2=0, or edge1=0 and edge2=1） does not result in a $\Delta KL$ exceeding the threshold, so this edge would be classified as "no contribution" and thus ablated. This is also why Nosing, with its focus on faithfulness, cannot guarantee circuit completeness.
>
> ### **Q4: It's not true that all circuits can be decomposed in OR, AND and ADDER (as defined)**
>
> Thank you for your question. We have addressed this issue in detail in our **response to Reviewer khCH’s Q1** (The **second reviewer in this paper**). In brief, the three gates are **not high-level constructs based on intuition**; rather, they form **a complete set that spans the intersection of Ns and Dn**, and other potential gates fall outside the scope of the Ns and Dn framework. Moreover, we will **include the potential gates in the limitation sections in the revised version.**
>
> ### **Q5: In line 103, these two papers are missing for linear estimation: EAP-IG and AtP. Both of these are reasonably cited elsewhere, especially EAP-IG. They address some of the issues of EAP/AtP for circuit attribution.**
>
> Thank you for your suggestions and we will add these to preliminaries in revised version!
>
> ### **Q6: In section 4.1, how do you choose the C_AND, C_OR and C_ADDER circuit parts? Using which Dn and Ns algorithm specifically? On how many random runs?**
>
> Thank you for your question. The notion of intersection in Section 4.1 refers to a **general theoretical framework** we propose, which is compatible with **any** specific implementation of either Ns or Dn algorithms. The detailed procedural steps for these implementations are provided in Appendix D.
>
> ### **Q7: How do you choose the 1 and 2 edges to remove for Figure 3? Randomly, you average over all sets of 1 edges and sets of 2 edges, ... ?**
>
> Thank you for your question. First, we filtered out the receiver nodes that contain more than 1 edge. For these nodes, we randomly ablated 1 or 2 edges and recorded the results. This process was performed 30 times on each node (we detailed these in line202-209). The final results presented in Figure 3 (box plot) correspond to the statistical values of all the experiments involving 1 edge and 2 edge ablations.
>
> ### **Q8: I think the "granularity alignment" concept from Figure 2 and Lines 222-233 needs more explanation. Why is Figure 2 a good picture of what's really happening in the circuit? The misalignment scores from Appendix E, they're completely separate for OR and AND; from figure 2 I thought misalignment was a property of the ADDER intersection? Very confused here.**
>
> Thank you for your question. We have addressed this issue in detail in our **response to Reviewer xbit’s Q1** (The **third reviewer in this paper**). In brief, by analyzing the trade-off between the misalignment scores of AND and OR gates, we observed that the ratio of AND to OR gates approaches 1:1 when the sum of their misalignment scores is minimized. This empirical observation underlies the conclusion presented in our paper. We will **include these experimental results in the revised version**.
>
> ### **Q9: Completeness validation 1: removing the circuit**
>
> Thank you for your question. The differences between ACDC and Edge-Pruning are minimal, except for EAP. This is because EAP is the only algorithm that cannot discover some OR gates within Ns (see Table 1).
>
> Additionally, as mentioned in lines 118-128, the metrics proposed by Wang et al. (2023)[1] often lead to unreliable approximations due to insufficient sampling, a conclusion that has been confirmed by numerous existing studies[2][3]. Therefore, we adopted the completeness metric proposed in paper [2].
>
> Moreover, in our **response to Reviewer mrxZ’s Q1**—**Key Question 1: How can we demonstrate that the new evaluation metric of completeness performs better than the previous evaluation metric?** (The **fourth reviewer in this paper**)—we present experimental evidence showing that the completeness metric proposed by Wang et al. suffers from **high variance** due to **insufficient sampling**, leading to **unstable results**. In contrast, our newly proposed completeness metric **significantly reduces the variance**, thereby yielding more reliable and consistent evaluations.
>
> **reference:**
>
> *[1]. Interpretability in the wild: a circuit for indirect object identification in gpt-2 small, ICLR 2023*
>
> *[2]. Functional faithfulness in the wild: Circuit discovery with differentiable computation graph pruning arxiv 2024*
>
> *[3]. Sparse interventions in language models with differentiable masking, ACL 2022*
>
>
> ### **Q10: completeness verification 2: average Hamming distance**
>
> Thank you for your question. The main purpose of this experiment is to verify the circuit overlap under **multiple random circuit discovery experiments**. Because a circuit algorithm with completeness is capable of discovering all circuit paths, so the circuits obtained in multiple random seeds should remain highly consistent. In contrast, a circuit lacking completeness is more likely to find **different branches across multiple random seeds**, leading to inconsistency. Therefore, our Hamming distance measures the distance between **any two circuits** under random seed conditions.  In revised versions, we will separate Table 3 into two subtables. Due to page constraints, we can present the separated results in the upcoming discussion.
>
> ### **Q11: How are you supporting the claim that "the AND gates typically receive edges from different functions" vs "the OR gates receive edges from the same function"? What is a function here? Do you literally mean whether incoming edges are from MLPs ("same function") or attentions and MLPs ("different function")? That's a very strange thing to check, different MLPs and different attention heads will be computing different function, so I don't see why they're the same**
>
> Thank you for your question. In lines 302-303, we explained that "*each color represents one function in the IOI circuit.*" Referring to Figure 10(a), these functions correspond to components such as the Duplicate Token Head, Induction Head, S-Inhibition Head, and others in the IOI model.
>
> ### **Q12: Limitations that aren't actually limitations**
>
> Thank you for your question. Our intention is similar to your **Q1**. We believe that the key metric for circuits remains faithfulness, and our method does not show a significant improvement in faithfulness (on the contrary, it shows a noticeable improvement in completeness). Therefore, compared to existing work, it is difficult to demonstrate **"superiority" in terms of faithfulness**. However, as discussed in **Q1**, completeness indeed holds potential for some valuable application scenarios.
>
> ### **Q13: Minor awkward wording**
>
> Thank you for your suggestion, we will update “introduce” to “borrow”.

---

> ### Author Response · Authors · 2025-08-04
> **Supplementary results for Q10**
>
> In response to Q10, the reviewer suggested that Table 3 should be decomposed into the two original Hamming distances rather than presenting only their difference. Accordingly, we provide below the raw data corresponding to Table 3. Specifically, Table 3 is derived by taking the difference between the sampled results shown in the two tables below:
>
> Table 1 The Hamming distance of existing circuit （Ns）
>
> | #edges | IOI       | IOI       | IOI       |  GT       |  GT       |  GT       |  SA       |  SA       |  SA       |
> | ------ | ----------| ----------| ----------| ----------| ----------| ----------| ----------| ----------| ----------|
> | #edges | ACDC | EAP     | Edge-Pruning|  ACDC | EAP     | Edge-Pruning|ACDC | EAP     | Edge-Pruning|
> |   100  |9.8±1.2    |3.8±0.8    |12.9±2.0   |10.2±1.3   |3.5±0.7    |18.5±2.3   |8.6±0.9    |3.5±0.6    |17.2±2.8    |
> |   200  |17.6±2.4   |6.4±1.1    |33.2±4.3   |19.5±2.6   |6.3±1.0    |41.5±5.1   |15.4±1.9   |8.4±1.3    |44.6±5.3   |
> |   500  |38.6±4.9   |7.9±1.4    |73.9±7.6   |43.9±5.6   |8.1±1.3    |79.6±6.8   |36.7±4.3   |9.5±1.3    |89.4±8.6   |
> |   1000 |61.7±6.9   |9.8±1.6    |129.7±22.7 |69.8±7.9   |9.4±1.3    |133.5±19.5 |59.7±7.3   |12.1±1.6   |175.9±24.9  |
> |   2000 |157.6±25.4 |14.9±1.8   |289.4±45.2 |166.3±29.7 |15.6±2.3   |301.4±44.1 |144.9±21.9 |18.6±1.7   |367.9±66.2  |
> |   5000 |234.8±36.9 |23.7±1.9   |744.9±224.7|267.3±44.1 |26.7±2.1   |812.6±234.1|212.8±33.5 |28.7±2.2   |988.6±311.5 |
>
> Table 2 The Hamming distance of our circuit （Ns+Dn）
> | #edges | IOI       | IOI       | IOI       |  GT       |  GT       |  GT       |  SA       |  SA       |  SA       |
> | ------ | ----------| ----------| ----------| ----------| ----------| ----------| ----------| ----------| ----------|
> | #edges | ACDC | EAP     | Edge-Pruning|  ACDC | EAP     | Edge-Pruning|ACDC | EAP     | Edge-Pruning|
> |   100  |5.8±0.3    |2.9±0.1    |6.7±0.8    |4.5±0.2    |2.6±0.1    |6.2±0.6    |5.8±0.4    |1.9±0.1    |4.7±0.4    |
> |   200  |11.4±0.5   |4.1±0.3    |16.8±1.9   |11.6±0.4   |4.1±0.2    |18.4±1.8   |9.5±0.4    |4.6±0.3    |18.4±1.6    |
> |   500  |31.7±1.9   |4.5±0.3    |39.5±2.4   |32.9±1.7   |4.6±0.3    |42.5±2.4   |28.4±1.4   |4.9±0.2    |44.5±2.6    |
> |   1000 |39.4±2.8   |5.7±0.5    |45.9±2.9   |41.5±2.4   |5.6±0.4    |49.7±2.6   |35.8±2.1   |6.3±0.4    |56.7±2.8   |
> |   2000 |83.9±5.7   |7.1±0.8    |94.6±5.5   |86.7±4.9   |7.3±0.4    |104.7±5.1  |74.8±6.6   |8.2±0.4    |127.4±5.2  |
> |   5000 |105.9±4.3  |9.5±1.7    |247.1±11.5 |101.4±4.1  |9.9±1.5    |273.9±10.5 |89.6±4.5   |10.6±0.9   |287.6±10.9 |
>
> We first generated different circuit graphs using various random seeds. The Hamming distance is used to quantify the dissimilarity between any pair of circuit graphs. The tables report the mean Hamming distances computed across 30 different random seeds. A larger Hamming distance indicates a lower degree of overlap among the circuits, which in turn suggests reduced structural completeness.

---

> ### Comment · Reviewer_mnpW · 2025-08-06
>
> First of all I apologize for the incomplete sentence in my review. "grounds for readers to believe that the presented method is actually better than the existing ones" definitely includes completeness, not just faithfulness. I appreciate that you have measured the impact of completeness on downstream tasks like model unlearning, but what about the measurement of completeness itself? Wang et al. 2022 (IOI) propose a definition of completeness which you use in this paper (L118-124), but don't actually use to measure how complete the resulting circuit is. **A claim that a method improves completeness should be accompanied by an increase in the completeness metric: to knockout circuit and model simultaneously and measure equality of behavior. Or an argument that this metric does not measure completeness well.**
>
> About definitions of gates. I think it's misleading to present these gates as something that just exists in models and from which we derive algorithms, when actually it's the other way around: the gates come from the definition of Ns and Dn algorithms and what kind of gates are captured by each (as you write to **ftP7**).
>
>
> >  they form a complete set that spans the intersection of Ns and Dn, and other potential gates fall outside the scope of the Ns and Dn framework.
>
> That seems correct, but the framework is something you only just introduced. You can't go around claiming that all circuits decompose into these fundamental things, that turn out to be part of a framework that this paper created.
>
> Also, there might be other kinds of gates that are detected by *neither Ns nor Dn*. For example, what about majority voting gates?  Noising any one input does not change the output, and denoising any one input does not change the output either;  so neither Dn nor Ns algorithms detect it. I don't think this falls out of the Dn or Ns framework, it's the obvious remaining category of it.
>
> You need much more argument explicitly supporting it, if you want to claim that OR, AND and ADDER are a complete set of gates.
>
> The algorithm does seem to improve on completeness discovery, but given the large conceptual errors of the paper and rebuttal, I will be keeping my score.

---

> > ### Author Response · Authors · 2025-08-06
> >
> > Thank you for your response. Based on the consensus we have reached with the other four reviewers regarding these issues, we would like to respectfully correct some of your misconceptions.
> >
> > **1. Completeness Evaluation**
> >
> > We provide both theoretical and empirical evidence that the metric proposed by Wang et al. (hereafter referred to as the *old metric*)[1] is inferior to the new metric proposed by Yu et al. (2024)[2].
> >
> > **Theoretical Evidence**: In the work of Yu et al. (2024)[2], not only is the *new metric* introduced, but the **subsection titled “Functional Completeness” (page 4)** also provides a detailed explanation of the limitations inherent in the *old metric* proposed by Wang et al. Specifically, they state:
> >
> > > “However, calculating this metric directly is intractable, leading to the use of random sampling of subsets, which provides an unreliable approximation.”
> >
> > **Empirical Evidence**: In response to **Reviewer mrxZ’s Question 1**—specifically, the key question “*How can we demonstrate that the new evaluation metric performs better than the old evaluation metric?*”—we presented experimental results showing that the *old metric* suffers from excessive variance due to insufficient sampling. In contrast, the *new metric* significantly reduces this variance.
> >
> > Therefore, we adopted the *new metric* rather than the *old metric* in our manuscript, a decision we believe to be both reasonable and well-justified. If you could provide specific reasons for your disagreement with this choice, we would be sincerely grateful.
> >
> >
> > **2. Definition of Gates**
> >
> > We reiterate that *logical gates* are an intrinsic property of circuits, and are **not** defined by either Ns or Dn. This view is supported in both [1] and [3], which emphasize that OR gates arise naturally within circuits. These methods do not utilize the intersection of Ns and Dn; instead, they identify the OR gate solely by examining the effects of ablating different edges.Additionally, [4] introduces the concepts of both AND and OR gates, stating explicitly:
> >
> > > “These AND and OR structures can appear in real-world transformers as serial-dependent components (e.g. a later attention head depending on an earlier one) or parallel components (such as redundant backup attention heads).”
> >
> > Furthermore, they share our position that:
> >
> > > “Ns and Dn behave differently on different gates.”
> >
> > Thus, we believe our definition of gates is sound and theoretically grounded. As a simple analogy, in a classification task, the different classes are naturally occurring, and we can identify them using a well-trained classifier. Similarly, logical gates are also naturally existing properties, and we can identify them through the intersection of Ns and Dn. However, **this does not imply that logical gates are defined by Ns and Dn.** This conceptual framework has also been accepted by all four of the other reviewers; even those who initially expressed doubts ultimately agreed with it following our clarification.
> >
> >
> > **3. Other Gates**
> >
> > Regarding the additional gate examples you mentioned, you note that:
> >
> > > “so neither Dn nor Ns algorithms detect it,”
> >
> > and also state:
> >
> > > “I don't think this falls out of the Dn or Ns framework.”
> >
> > We find this reasoning to be inherently contradictory. If neither Dn nor Ns can detect such structures, then it is not possible to identify them through the intersection of circuits derived from Dn and Ns. Therefore, these gates lie outside the scope of our study.
> >
> > Moreover, we have demonstrated that *every edge in a circuit* can be categorized into one of the three gate types we define (see Figure 10 for complete circuit graphs). Consequently, we believe that our use of these three gate types to explain the *faithfulness* and *completeness* of circuits is both comprehensive and appropriate.
> >
> > **reference**
> >
> > *[1]. Interpretability in the Wild: a Circuit for Indirect Object Identification in GPT-2 small*
> >
> > *[2]. Functional Faithfulness in the Wild: Circuit Discovery with Differentiable Computation Graph Pruning*
> >
> > *[3]. Towards Automated Circuit Discovery for Mechanistic Interpretability*
> >
> > *[4]. How to use and interpret activation patching*

---

### Author Response · Authors · 2025-08-09
**Summary of Key Questions and Committed Revisions**

Thank you to all reviewers and the chair for their efforts during the discussion stage. To facilitate the next stage of decision-making, we have objectively summarized the main concerns raised by the reviewers and listed the revisions we have committed to (most of which have already been completed).

**Main Concerns:**

**Q1: Ambiguity in the definition of “circuit gate”** (from Reviewer *mnpW* Q2, Reviewer *ftp7* Q1)

We have adopted the suggestion from Reviewer *ftp7*. Specifically, we emphasize that the term *logical gate (state)* merely characterizes how the output of a function influences the final output when its inputs are either ablated (*state* = 0) or retained (*state* = 1), rather than describing the function itself as a logical function of its inputs. Accordingly, in our reply to Reviewer *ftp7*, we provided a revised version of Definition 1 to highlight the distinction between a node’s *activation state* and *activation value*. **This clarification was endorsed by Reviewers khCH and ftp7**.

**Q2: Existence of other types of gates** (from Reviewer *mnpW* Q4, *khCH* Q1, *ftp7* Q4)

The intersection of $N_s$ and $D_n$ in this work reveals only three types of gates: AND, OR, and ADDER. Other potential gates cannot be detected through the $N_s$–$D_n$ intersection and thus fall beyond the scope of the present study. We will explicitly note this possibility in the *Limitations* section. **This explanation was accepted by Reviewers khCH and ftp7.**

**Q3: Significance/advantage of improving circuit completeness** (from Reviewer *mnpW* Q1, *xbit* Q2, *mrxZ* Q1)

Through a model unlearning task, we demonstrate that improving circuit completeness enables a more thorough removal of the mechanisms related to the target concept. The specific results are provided in our reply to Reviewer mrxZ Q1. **This explanation was acknowledged by Reviewers xbit and mrxZ.**

**Q4: Rationale for adopting the new completeness metric instead of the previous one** (from Reviewer *mnpW* Q9, *mrxZ* Q1)

Experimental results confirm that the previous metric suffers from high evaluation variance due to insufficient sampling, whereas the new metric exhibits more stable performance. We also cite explicit statements from prior work indicating that the previous metric’s estimates are compromised by inadequate sampling. The relevant results are presented in our reply to Reviewer mrxZ Q1. **This explanation was approved by Reviewer mrxZ.**

**Q5: Why the optimal granularity ratio between AND and OR gates is 1:1** (from Reviewer *mnpW* Q6, *xbit* Q1)

Our experiments show that when the ratio of the number of AND edges to OR edges is approximately 1:1, the *misalignment score* for both gate types jointly attains its optimal value. The results are detailed in our reply to Reviewer xbit Q1. **This explanation was accepted by Reviewer xbit.**

---

**Committed Revisions (already completed):**

* **M1: Method**

  We rewrote the definition of “circuit gate” to stress the distinction between *activation state* and *activation value*. The revised version is presented in the discussion with Reviewer ftp7.

* **M2: Limitation**

  We added a statement acknowledging the possible existence of other gates and explained that their identification lies beyond the exploration capacity of the $N_s$–$D_n$ framework.

* **M3: Broader Impact**

  We included results from the model unlearning task, showing that circuit completeness can facilitate complete forgetting of target mechanisms. The findings are provided in our reply to Reviewer mrxZ Q1.

* **M4: Appendix**

  We added a comparison between the old and new completeness metrics (results in mrxZ Q1), a proof of the NP problem (ftp7 Q3), and the detailed results on optimal granularity (xbit Q1).

* **M5: Experiments**

  We will refine Figure 4 for clearer presentation and supplement it with an evaluation of the $D_n$ algorithm’s performance on completeness (results in khCH Q8).

We extend our sincere gratitude to Reviewers khCH, xbit, mrxZ, and ftp7 for their constructive discussions and positive score.

---

### Note · Authors · 2025-08-12

Thank you to Chair and reviewers for the time and effort evaluating our work. Prior to this response, we provided a **General Response** titled *“Summary of Key Questions and Committed Revisions”*, which enumerates the primary issues discussed with the reviewers and the revisions we have already implemented. Here, we supplement that summary with further clarifications.

### **Contributions:**
The main contribution of this work is the systematic introduction of three types of logical gates—**AND**, **OR**, and **ADDER**— for representing circuits derived through *Noising* or *Denoising* procedures. This framework reveals the relationship between the effect on an output and the state of a receiver node when the corresponding sender node is either ablated or retained, analogous to the truth tables of AND, OR, and ADDER gates in digital logic.

Building upon this, we use logical gates to analyze the **minimal requirements for achieving *faithfulness* and *completeness* in circuits**. Furthermore, we **introduce a combined *Noising + Denoising* approach** to recover circuits that are *logically sound*. Through this analysis, we uncover systematic differences among AND, OR, and ADDER gates in terms of their **abundance, patterns of effect variation, and functional roles**.

### **Reviewer stance:**
Following discussions with five reviewers, all concerns raised by the four reviewers (**khCH**, **xbit**, **mrxZ**, and **ftp7**) were satisfactorily addressed, and they **committed to providing positive scores** (4, 4, 5, 4, respectively). However, reviewer **mnpW** maintained a **negative assessment** (2) due to significant misunderstandings of our work. **We believe the paper remains worthy of acceptance, as the issues raised by mnpW were also raised by other reviewers, discussed in detail, and ultimately affirmed by them. Moreover, mnpW did not provide sufficient objective justification during the discussion phase to refute our conclusions.**

### **Committed revisions** *(most of which have already been completed and are documented in the discussion)*:
We committed to revising the definitions in the *Method* section, as well as adding corresponding discussions and experimental results in the *Limitations*, *Appendix*, and *Experiments* sections. **The specific details of these changes are available in the *“Summary of Key Questions and Committed Revisions”* General Response.**

Finally, we again express our thanks to reviewers and the Chair.

---

### Decision · Program_Chairs · 2025-09-17

**Decision:**

Accept (poster)

**Comment:**

This paper reframes circuit discovery in mechanistic interpretability by introducing three logical gate types as abstractions of how components behave under noising and denoising interventions. The authors argue that incompleteness in existing methods largely stems from failure to detect OR gates, and propose a combined Ns+Dn approach that allows circuits to be recovered with greater completeness. Empirical studies on GPT-2 small across canonical interpretability tasks show that the Ns+Dn framework yields circuits that are more complete and less random across seeds, while maintaining comparable faithfulness. The work also explores the distribution and roles of these gates, suggesting OR gates capture redundancy, AND gates combine distinct functions, and ADDER gates provide additive contributions.

The reviewers were divided. One reviewer (mnpW) maintained a strong rejection, citing conceptual flaws in presenting AND/OR/ADDER as "fundamental" and objecting to the incompleteness of the definitions and completeness metrics. However, the other reviewers judged that the clarifications and revisions provided by the authors during the rebuttal period resolved most concerns. In particular, khCH and ftP7 initially shared mnpW’s confusion about definitions but accepted the authors' clarification that gates describe ablation-induced behavior rather than literal logical functions. Similarly, reviewers xbit and mrxZ appreciated the new completeness analysis and the demonstration of practical value in model unlearning. Across the discussion, consensus formed that the work is conceptually interesting, well-motivated, and empirically consistent, though its claims should be carefully delimited.

Strengths of the paper include a clear conceptual framing of incompleteness, a simple but effective Ns+Dn intervention strategy, extensive empirical validation across multiple circuit discovery baselines, and thoughtful exploratory analysis of the roles of different gate types. The rebuttal further strengthened the case by demonstrating that greater completeness can enable more thorough unlearning of harmful circuits.

The weaknesses remain worth noting. The framework is incremental in technical depth, since Ns+Dn is a straightforward combination of existing interventions. The definitions of gates required multiple rounds of clarification, and some readers may still find the logical analogy overextended. The empirical studies are limited to GPT-2 small, and faithfulness is not improved relative to prior methods, which may limit adoption by practitioners focused on that metric. Finally, ADDER gates function as a broad "catch-all" category and their interpretability value is less compelling than that of OR or AND gates.

Overall, the clarifications and revisions provided by the authors during the disuccsion period  and the potential impact of framing circuit completeness through the lens of logical gates makes me to recommend a weak acceptance (i.e., poster). The main recommendation to the authors is to revise the presentation carefully by making explicit that gate definitions describe ablation states rather than intrinsic node functions and clearly delimiting the scope to circuits derived from Ns and Dn, while acknowledging that other gate types may exist. Finally, the authors should improve the readability of figures and include quantitative tables to support claims.